# THE ROLE OF COUNTERFACTUAL EXPLANATIONS IN MODEL EXTRACTION ATTACKS

## ABSTRACT

Counterfactuals provide guidance on achieving a favorable outcome from a model, with minimum input perturbation. However, counterfactuals can also be exploited to leak information about the underlying model, causing privacy concerns. Prior work shows that one can query for counterfactuals with several input instances and train a surrogate model using all the queries and their counterfactuals. In this work, we analyze how model extraction attacks can be improved by further leveraging the fact that *the counterfactuals also lie quite close to the decision boundary*. Using polytope theory, we derive a novel theoretical relationship between the error in model approximation and the number of queries, when the queries exactly return the "closest" counterfactual. Noting the practicalities of counterfactual generation, we also provide additional theoretical guarantees leveraging Lipschitz continuity, that hold when the counterfactuals are reasonably close but may not be the closest ones. Our theoretical results help us arrive at a simple strategy for model extraction, which includes a loss function that treats counterfactuals differently than ordinary instances. Our approach also alleviates the related problem of "decision boundary shift". Experimental results demonstrate the performance of our strategy on synthetic data as well as popular real-world tabular datasets.

## 1 INTRODUCTION

As machine learning becomes ubiquitous in consequential decision-making, there is an increasing interest in post-hoc explanation methods (Slack et al., 2021; Han et al., 2022). Post-hoc explanations provide insights on the decision-making of complex models after they have already been trained. Among post-hoc explanation methods, counterfactual explanations (Wachter et al., 2017) provide a unique functionality: They provide guidance on how to change the input features to achieve a more favorable outcome, e.g., increase your income by 10K to qualify for the loan. Given an input instance, a counterfactual explanation (also called *a counterfactual*) is another instance that belongs to a different output class. Typically, the counterfactual is selected based on certain desirable criteria such as proximity to the original instance, change in as few features as possible, etc. Counterfactuals are simple enough for the users to understand, enabling a way to provide feedback, and build trust.

Despite the many advantages they offer, counterfactuals also have the potential to leak sensitive information about the underlying model. It can become a serious threat, especially if users can query the model, as in Machine Learning as a Service (MLaaS) platforms (Gong et al., 2021; Tramèr et al., 2016). MLaaS has become a popular alternative to learning models on-site. The services are usually monetized, and the users are typically charged for the number of queries (Gong et al., 2020; Juuti et al., 2019). Preserving the privacy of these models is a huge concern for the operators of such services because by strategically querying the model, an adversary may be able to "steal" the model (Wang et al., 2022; Gong et al., 2021; Tramèr et al., 2016; Pal et al., 2020; Juuti et al., 2019). Stealing usually involves training a surrogate model to provide similar predictions as the target model, a practice also referred to as *model extraction*. In this work, our main question is: *How to provide theoretical guarantees on model extraction attacks using counterfactual explanations?*.

Using counterfactuals for model extraction has received limited attention. One existing method is to treat counterfactuals as ordinary labeled points and use them for training the surrogate model (Aïvodji et al., 2020). While this may work for a well-balanced query dataset with queries from the two classes lying roughly equidistant to the decision boundary, it is not the same for unbal-

anced datasets. The surrogate decision boundary might not always overlap with that of the target model (see Figure 1), a problem also referred to as *a decision boundary shift* (Wang et al., 2022).

This is a result of the learning process where the boundary is typically kept as far as possible from the training examples (margin) to achieve better generalization (Shokri et al., 2021). This issue is aggravated when the system provides only *one-sided counterfactuals*, i.e., counterfactuals only for queries with unfavorable predictions which is a common use case, e.g., only for rejected applicants in a loan application. This is because the counterfactuals are typically quite close to the decision boundary. *Hence, when treated as ordinary labeled points for training the surrogate model, one-sided counterfactuals make the dataset unbalanced in terms of the distance from the decision boundary.*

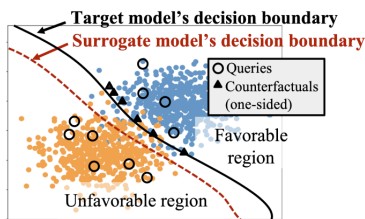

Figure 1: Decision boundary shift.

In this work, we analyze how model extraction attacks can be improved by leveraging the fact that the counterfactuals are close to the decision boundary, and demonstrate strategies that alleviate the decision-boundary-shift issue. We provide novel theoretical guarantees for the attacks, addressing an important knowledge gap in the existing literature. In contrast to existing attacks Aïvodji et al. (2020) and Wang et al. (2022) which require the system to provide counterfactuals for queries from both sides of the decision boundary, our methods require only one-sided counterfactuals (from one side of the decision boundary). Our contributions can be listed concisely as follows:

- **Guarantees on model approximation using closest counterfactuals:** We first demonstrate how the closest counterfactuals can provide a linear approximation of the decision boundary around the counterfactuals (Theorem 1). Then, we use results from the polytope theory to analyze the goodness of this linear approximation. In this regard, we theoretically characterize the relationship between the average extent of agreement of the surrogate model (obtained using linear approximations) with the target model, and the number of queries required (Theorem 2).

- **An attack based on the Lipschitz continuity of models:** Noting the difficulties associated with generating the closest counterfactuals, we focus on an alternative strategy where we exploit only the fact that counterfactuals lie reasonably close to the decision boundary, but need not be exactly the closest. Accordingly, we observe that the Lipschitz continuity of the models constricts deviations of the prediction probabilities around matching points (Theorem 3). To be precise, if $m_1$ and $m_2$ are two $\gamma-$Lipschitz continuous classifiers and $c$ is a matching point, i.e., an input instance such that $m_1(c) = m_2(c)$, then for any $x$ in the input space, $m_1(x)$ and $m_2(x)$ differ at most by $2\gamma||x - c||_2$. This observation leads to a new way of utilizing counterfactuals in model extraction. We note that by forcing the surrogate model to match the prediction probabilities of the target model at a sufficiently large number of points along the decision boundary, we can force the decision boundaries of the target and surrogate models to overlap significantly. We further note that this can be achieved with counterfactuals, as they lie closer to the target decision boundary where the prediction probability of the target model is known to be 0.5.

- **An analysis of the attack based on the Lipschitz continuity:** We also evaluate the query complexity of the aforementioned strategy for target models with a convex decision boundary under the assumption of closest counterfactuals (Theorem 4). We further look into the case of monotonic models, where a reduction in the query complexity is observed (Corollary 1).

- **A novel loss function and empirical validation:** We implement the aforementioned strategy through a novel loss function (Equation 5) for neural-network-based classifiers whose decision boundaries are not necessarily convex. Our proposed loss function forces the surrogate models to predict a pre-determined probability for counterfactuals rather than treating them as ordinary labeled instances. We conduct experiments on both synthetic datasets, as well as, four real-world datasets, namely, Adult Income (Becker & Kohavi, 1996), COMPAS (Angwin et al., 2016), DCCC (Yeh, 2016), and HELOC (FICO, 2018). Our strategy outperforms the attack by Aïvodji et al. (2020) over all these datasets (Section 4) using one-sided counterfactuals, i.e., only queries from the unfavorable side of the decision boundary.

***Related Works:*** A plethora of counterfactual-generating mechanisms has been suggested in existing literature (Guidotti, 2022; Verma et al., 2022; Karimi et al., 2022). In addition to the proximity to the original instance, these mechanisms focus on properties such as plausibility (Karimi et al., 2020), diversity (Mothilal et al., 2020), sparsity (Dhurandhar et al., 2018), and other pre-defined constraints

(Deutch & Frost, 2019). Despite being useful as a post-hoc explanation method, counterfactuals may leak sensitive information about the underlying dataset or the model. Related works that focus on leaking information about the dataset from counterfactual explanations include membership inference attacks (Pawelczyk et al., 2023) and explanation-linkage attacks (Goethals et al., 2023). Previously, Shokri et al. (2021) looked into membership inference from other types of explanations, e.g., feature-based. Instead, we focus on model extraction from counterfactual explanations.

Model extraction attacks (without counterfactuals) have been the topic of a wide array of studies (see surveys Gong et al. (2020) and Oliynyk et al. (2023)). Various mechanisms such as model inversion (Gong et al., 2021), equation solving (Tramèr et al., 2016), as well as active learning have been considered (Pal et al., 2020). Milli et al. (2019) looks into model reconstruction using other types of explanations, e.g., gradient-based. Yadav et al. (2023) explore algorithmic auditing using counterfactual explanations, focusing on linear classifiers and decision trees. Using counterfactual explanations for model extraction has received limited attention, with the notable exception of Aïvodji et al. (2020) and Wang et al. (2022). Aïvodji et al. (2020) suggest using counterfactuals as ordinary labeled examples while training the surrogate model. This suffers from the issue of decision boundary shift, particularly for unbalanced query datasets (one-sided counterfactuals). Wang et al. (2022) is a significant contribution which introduces a clever strategy of further querying for the counterfactual of the counterfactual. Both these methods require the system to provide counterfactuals for queries from both sides of the decision boundary. Nevertheless, a user with a favorable decision may not usually require a counterfactual explanation, and hence a system providing one-sided counterfactuals might be more common, wherein lies our significance. While model extraction attacks (without counterfactuals) have received interest from a theoretical perspective (Tramèr et al., 2016; Papernot et al., 2017; Milli et al., 2019), model extraction attacks involving counterfactual explanations lack such a theoretical understanding. Our work proposes and theoretically analyzes model extraction attacks by explicitly utilizing the fact that the counterfactuals are close to the decision boundary. We provide novel performance guarantees in terms of query complexity for the proposed attacks, leveraging polytope theory, and also address the decision-boundary shift issue.

## 2 PRELIMINARIES

**Notations:** We consider machine learning models $m : \mathbb{R}^d \to [0, 1]$ for binary classification that take an input value $x \in \mathbb{R}^d$ and output a probability between $0$ and $1$. The final predicted class is denoted by $\lfloor m(\boldsymbol{x}) \rceil \in \{0, 1\}$ which is obtained by thresholding the output probability at $0.5$ as follows: $\lfloor m(\boldsymbol{x}) \rceil = \mathbb{1}[m(\boldsymbol{x}) \geq 0.5]$ where $\mathbb{1}[\cdot]$ denotes the indicator function. Throughout the paper, we denote the output probability by $m(\boldsymbol{x})$ and the corresponding thresholded output by $\lfloor m(\boldsymbol{x}) \rceil$. Consequently, the decision boundary of the model $m$ is the $(d-1)$-dimensional hypersurface (generalization of surface in higher dimensions; see Definition 4) in the input space, given by $\partial \mathbb{M} = \{\boldsymbol{x} : m(\boldsymbol{x}) = 0.5\}$. We call the region where $\lfloor m(\boldsymbol{x}) \rceil = 1$ as the *favorable region* and the region where $\lfloor m(\boldsymbol{x}) \rceil = 0$ as the *unfavorable region*. We always state the convexity/concavity of the decision boundary with respect to the favorable region (i.e., the decision boundary is convex if the set $\mathbb{M} = \{\boldsymbol{x} \in \mathbb{R}^d : \lfloor m(\boldsymbol{x}) \rceil = 1\}$ is convex). We assume that upon knowing the range of values for each feature, the $d-$dimensional input space can be normalized so that the inputs lie within the set $[0, 1]^d$ (the $d-$dimensional unit hypercube), as is common in literature (Liu et al., 2020; Tramèr et al., 2016; Hamman et al., 2023; Black et al., 2022). We denote by $g_m$, the counterfactual generating mechanism corresponding to the model $m$, which is defined next.

**Definition 1** (Counterfactual Generating Mechanism). *Given a cost function $c : [0, 1]^d \times [0, 1]^d \to \mathbb{R}_0^+$ for measuring the quality of a counterfactual, and a model $m$, the corresponding counterfactual generating mechanism is the mapping $g_m : [0, 1]^d \to [0, 1]^d$ specified as*

$$g_m(\boldsymbol{x}) = \underset{\substack{\boldsymbol{w} \in [0,1]^d \\ \lfloor m(\boldsymbol{x}) \rceil \neq \lfloor m(\boldsymbol{w}) \rceil}}{\arg \min} c(\boldsymbol{x}, \boldsymbol{w}). \tag{1}$$

The cost $c(\boldsymbol{x}, \boldsymbol{w})$ is selected based on specific desirable criteria, e.g., $c(\boldsymbol{x}, \boldsymbol{w}) = ||\boldsymbol{x} - \boldsymbol{w}||_p$, with $|| \cdot ||_p$ denoting the $L_p$-norm. Specifically, $p = 2$ leads to the following definition of the *closest counterfactual* (Wachter et al., 2017; Laugel et al., 2017; Mothilal et al., 2020).

**Definition 2** (Closest Counterfactual). *When $c(\boldsymbol{x}, \boldsymbol{w}) \equiv ||\boldsymbol{x} - \boldsymbol{w}||_2$, the resulting counterfactual generated using $g_m$ as per Definition 1 is called the "closest counterfactual."*

**Problem Setting:** Our problem setting involves a target model $m$ which is pre-trained and assumed to be hosted on a MLaaS platform (see Figure 2). Users can query it with a set of input instances $\mathbb{D} \subseteq [0,1]^d$ through an Application Programming Interface (API). The API will provide the users with the set of predictions, i.e., $\{\lfloor m(\boldsymbol{x}) \rceil : \boldsymbol{x} \in \mathbb{D}\}$, and a set of *one-sided* counterfactuals for the instances whose predicted class is 0, i.e., $\{g_m(\boldsymbol{x}) : \boldsymbol{x} \in \mathbb{D}, \lfloor m(\boldsymbol{x}) \rceil = 0\}$. Note that, by the definition of a counterfactual, $\lfloor m(g_m(\boldsymbol{x})) \rceil = 1$ for all $\boldsymbol{x}$ with $\lfloor m(\boldsymbol{x}) \rceil = 0$. An adversary, while appearing to be a normal user, can query with an attack dataset $\mathbb{D}_{\text{attack}} \subseteq [0,1]^d$ and use the returned labels and counterfactuals to train their

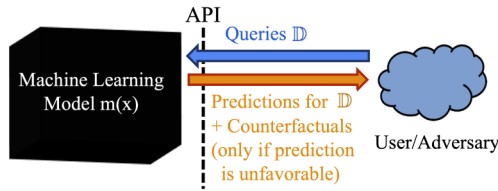

Figure 2: Problem setting.

own *surrogate model* $\tilde{m}$. The **goal** of the adversary is to achieve a certain level of performance with as few queries as possible. In this work, we use *fidelity* as our performance metric for model extraction[1]. Next, we provide the definition of fidelity over a reference set of input instances ($\mathbb{D}_{\text{ref}}$).

**Definition 3** (Fidelity (Aïvodji et al., 2020)). *With respect to a given target model $m$ and a reference dataset $\mathbb{D}_{ref} \subseteq [0,1]^d$, the fidelity of a surrogate model $\tilde{m}$ is given by*

$$\text{Fid}_{m,\mathbb{D}_{ref}}(\tilde{m}) = \frac{1}{|\mathbb{D}_{ref}|} \sum_{\boldsymbol{x} \in \mathbb{D}_{ref}} \mathbb{1}\left[\lfloor m(\boldsymbol{x}) \rceil = \lfloor \tilde{m}(\boldsymbol{x}) \rceil\right].$$

**Background on Geometry of Decision Boundaries:** Our theoretical analysis employs arguments based on the geometry of the involved models' decision boundaries. We assume the decision boundaries are hypersurfaces. A hypersurface is a generalization of a surface into higher dimensions, e.g., a line or a curve in a 2-dimensional space, a surface in a 3-dimensional space, etc.

**Definition 4** (Hypersurface, Lee (2009)). *A hypersurface is a $(d-1)$-dimensional sub-manifold embedded in $\mathbb{R}^d$, which can also be denoted by a single implicit equation $\mathcal{S}(\boldsymbol{x}) = 0$ where $\boldsymbol{x} \in \mathbb{R}^d$.*

We focus on the properties of hypersurfaces which are "touching" each other, as defined next.

**Definition 5** (Touching Hypersurfaces). *Let $\mathcal{S}(\boldsymbol{x}) = 0$ and $\mathcal{T}(\boldsymbol{x}) = 0$ denote two differentiable hypersurfaces in $\mathbb{R}^d$. $\mathcal{S}(\boldsymbol{x}) = 0$ and $\mathcal{T}(\boldsymbol{x}) = 0$ are said to be touching each other at the point $\boldsymbol{w}$ if and only if $\mathcal{S}(\boldsymbol{w}) = \mathcal{T}(\boldsymbol{w}) = 0$, and there exists a non-empty neighborhood $\mathcal{B}_{\boldsymbol{w}}$ around $\boldsymbol{w}$, such that $\forall \boldsymbol{x} \in \mathcal{B}_{\boldsymbol{w}}$ with $\mathcal{S}(\boldsymbol{x}) = 0$ and $\boldsymbol{x} \neq \boldsymbol{w}$, only one of $\mathcal{T}(\boldsymbol{x}) > 0$ or $\mathcal{T}(\boldsymbol{x}) < 0$ holds. (i.e., within $\mathcal{B}_{\boldsymbol{w}}, \mathcal{S}(\boldsymbol{x}) = 0$ and $\mathcal{T}(\boldsymbol{x}) = 0$ lie on the same side of each other).*

Next, in Lemma 1 we show that touching hypersurfaces share a common tangent hyperplane at their point of contact. This result is instrumental in exploiting the closest counterfactuals in a model extraction attack. The proof is deferred to Appendix A.1.

**Lemma 1.** *Let $\mathcal{S}(\boldsymbol{x}) = 0$ and $\mathcal{T}(\boldsymbol{x}) = 0$ denote two differentiable hypersurfaces in $\mathbb{R}^d$, touching each other at point $\boldsymbol{w}$. Then, $\mathcal{S}(\boldsymbol{x}) = 0$ and $\mathcal{T}(\boldsymbol{x}) = 0$ have a common tangent hyperplane at $\boldsymbol{w}$.*

**Properties of Classifiers:** We now introduce two properties of neural-network-based classifiers, namely Lipschitz continuity and monotonicity, which are often encountered in related works (Bartlett et al., 2017; Gouk et al., 2021; Pauli et al., 2021; Hamman et al., 2023; Liu et al., 2020; Marques-Silva et al., 2021). These properties are commonly observed in practice and can be exploited in perfecting model extraction attacks.

**Definition 6** (Lipschitz Continuity). *A model $m$ is $\gamma-$Lipschitz continuous if and only if*

$$|m(\boldsymbol{x}_1) - m(\boldsymbol{x}_2)| \leq \gamma ||\boldsymbol{x}_1 - \boldsymbol{x}_2||_2 \tag{2}$$

*for all $\boldsymbol{x}_1, \boldsymbol{x}_2 \in [0,1]^d$, with $\gamma \in \mathbb{R}_0^+$ and $|\cdot|$ denoting the absolute value.*

**Definition 7** (Monotonicity in a Feature). *A model $m(\boldsymbol{x})$ is monotonic in feature $i$ if for all input vectors $\boldsymbol{x}, \boldsymbol{x}' \in [0,1]^d$ such that $x_j \geq x'_j$ for $j = i$ and $x_j = x'_j$ for $j \neq i$, we have $m(\boldsymbol{x}) \geq m(\boldsymbol{x}')$.*

---

[1]The performance can be evaluated using either accuracy or fidelity (Jagielski et al., 2020). Accuracy is a measure of how well the surrogate model can predict the true labels, over the data manifold of interest. Fidelity measures the agreement between labels predicted by the surrogate and the target models. While attacks based on both measures have been proposed in literature, fidelity-based attacks have been deemed more useful as a first step in designing and mounting future attacks (Jagielski et al., 2020; Papernot et al., 2017).

## 3 MAIN RESULTS

### 3.1 GUARANTEES ON MODEL APPROXIMATION USING CLOSEST COUNTERFACTUALS

We first start out with demonstrating how the closest counterfactuals provide a linear approximation of *any* decision boundary. Prior work (Yadav et al., 2023) show that for linear models, the line joining a query instance $x$ and the closest counterfactual $w(=g_m(x))$ is perpendicular to the linear decision boundary. We generalize this observation to any differentiable decision boundary, not necessarily linear, as presented in the following theorem.

**Theorem 1.** *Let $\mathcal{S}$ denote the decision boundary of a classifier and $x \in [0,1]^d$ be any point that is not on $\mathcal{S}$. Then, the line joining $x$ and its closest counterfactual $w$ is perpendicular to $\mathcal{S}$ at $w$.*

The proof follows by showing that the $d$-dimensional ball with radius $||x - w||_2$ touches (as in Definition 5) $\mathcal{S}$ at $w$, and invoking Lemma 1. For details see Appendix A.1.

As a direct consequence of Theorem 1, an adversary may query the system and calculate tangent hyperplanes of the decision boundary drawn at the counterfactual instances. This leads to a linear approximation of the decision boundary (see Figure 3).

If the decision boundary is convex from the unfavorable side, i.e., the region where $\lfloor m(x) \rceil = 0$, such an approximation will provide a set of supporting hyperplanes. *The intersection of these supporting hyperplanes will provide a circumscribing polytope approximation of the decision boundary.* We show that the fidelity of such an approximation, evaluated over uniformly distributed input instances, tends to 1 for large $n$.

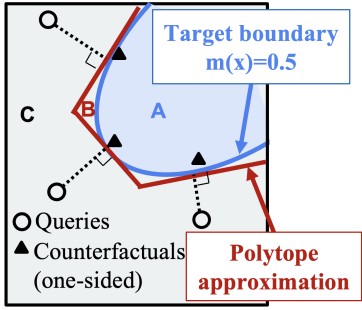

Figure 3: Polytope approximation of a convex decision boundary.

**Theorem 2.** *Let $m$ be the target binary classifier whose decision boundary is convex (i.e., the set $\{x \in [0,1]^d : \lfloor m(x) \rceil = 1\}$ is convex) and has a continuous second derivative. Denote by $\tilde{M}_n$, the polytope approximation of $m$ constructed with $n$ supporting hyperplanes obtained through i.i.d. queries. Assume that the fidelity is evaluated with respect to a $\mathbb{D}_{ref}$ which is uniformly distributed over $[0,1]^d$. Then, when $n \to \infty$ the expected fidelity of $\tilde{M}_n$ is given by*

$$\mathbb{E}\left[\mathrm{Fid}_{m,\mathbb{D}_{ref}}(\tilde{M}_n)\right] = 1 - \epsilon \tag{3}$$

*where $\epsilon \sim \mathcal{O}\left(n^{-\frac{2}{d-1}}\right)$ and the expectation is over both $\tilde{M}_n$ and $\mathbb{D}_{ref}$.*

The proof utilizes a result from polytope theory (Böröczky Jr & Reitzner, 2004) which provides a complexity result on volume-approximating smooth convex sets by random polytopes. The proof involves observing that the volume of the overlapping decision regions of $m$ and $\tilde{M}_n$ (for example, regions A and C in Figure 3) translates to the expected fidelity when evaluated under a uniformly distributed $\mathbb{D}_{ref}$. Appendix A.2 provides the detailed steps.

**Remark 1** (Relaxing the Convexity Assumption). *Even though Theorem 2 assumes convexity of the decision boundary for analytical tractability, the attack can be extended to a concave decision boundary. This is because the closest counterfactual will always lead to a tangent hyperplane irrespective of convexity and now the rejected region can be seen as the intersection of these half-spaces (Theorem 1 does not assume convexity). However, it is worth noting that approximating a concave decision boundary is, in general, more difficult than approximating a convex region. To obtain equally-spaced out tangent hyperplanes on the decision boundary, a concave region will require a much denser set of query points (see Figure 4) due to the inverse effect of length contraction discussed in Aleksandrov (1967, Chapter III Lemma 2). Furthermore, approximating a decision boundary which is neither convex nor concave is much more challenging as the decision regions can no longer be approximated as intersections of half-spaces. This*

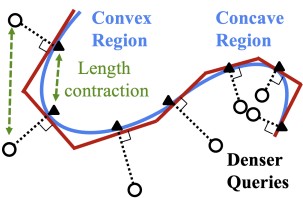

Figure 4: Approximating concave regions need denser queries than convex regions.

*motivates us to propose an attack that does not depend on convexity assumption leveraging Lipschitz continuity, as discussed next. Experiments indicate that the query complexity of this Lipschitz-based attack is upper-bounded by the result in Theorem 2 (see Figure 16).*

## 3.2 An attack based on the Lipschitz continuity of the models

Generating the closest counterfactuals, which are used in the attack discussed in Section 3.1, is often a challenging task. Hence, in this Section, we propose an alternative strategy that depends only on the fact that counterfactuals lie closer to the decision boundary but need not be the closest. Here, we consider the difference of the model output probabilities (before thresholding) as the measure of similarity between the target and surrogate models. This metric forces the decision boundaries of the two models to be overlapped, and hence, will act as a proxy to the fidelity. The proposed attack is based on the Lipschitz continuity of the models involved.

In the following theorem, we show that the difference of the model outputs corresponding to a given input instance can be bounded by having a point with matching outputs in the affinity of that instance. This is the key observation in devising the new attack.

**Theorem 3.** *Suppose the target ($m(\boldsymbol{x})$) and the surrogate ($\tilde{m}(\boldsymbol{x})$) models are $\gamma$-Lipschitz continuous. Assume $m(\boldsymbol{w}) = \tilde{m}(\boldsymbol{w})$ for some $\boldsymbol{w} \in [0, 1]^d$. Then, for any $\boldsymbol{x} \in [0, 1]^d$, the difference between the outputs of the two models is bounded from above as follows;*

$$|\tilde{m}(\boldsymbol{x}) - m(\boldsymbol{x})| \leq 2\gamma ||\boldsymbol{x} - \boldsymbol{w}||_2. \tag{4}$$

The proof is presented in Appendix A.3. Usually, a smaller Lipschitz constant is indicative of a higher generalizability of a machine learning model (Gouk et al., 2021; Pauli et al., 2021). Therefore, it may be reasonable to assume $\gamma$ above is relatively small.

**Remark 2** (Local Lipschitz Continuity). *It is noteworthy that if a well-spread set of points $\mathbb{W} = \{\boldsymbol{w}_i, i = 1, \dots, N\}$ which satisfies $m(\boldsymbol{w}_i) = \tilde{m}(\boldsymbol{w}_i)$ for $i = 1, \dots, N$ is available over some region $\mathbb{A} \subseteq [0, 1]^d$, then local $\gamma-$Lipschitz continuity of $m$ and $\tilde{m}$ in the locality of $\boldsymbol{w}_i$'s is sufficient to ensure the conditions of Theorem 3 for all $\boldsymbol{x} \in \mathbb{A}$.*

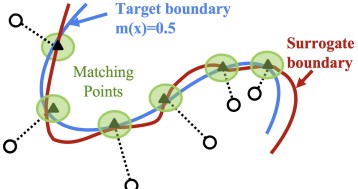

Figure 5: An attack based on the Lipschitz continuity of target and surrogate models.

**Proposed attack:** Theorem 3 provides the motivation for a novel model extraction strategy. Let $\boldsymbol{w}$ be a generic instance of a set of counterfactuals generated by a certain query. Recall that $\partial\mathbb{M}$ denotes the decision boundary of $m$. As implied by the theorem, for any $\boldsymbol{x} \in \partial\mathbb{M}$, the deviation of the surrogate model output from the target model output is bounded above by a multiple of the distance between $\boldsymbol{x}$ and the counterfactual that lies closest to it, given that all the counterfactuals satisfy $m(\boldsymbol{w}) = \tilde{m}(\boldsymbol{w})$. Knowing that $m(\boldsymbol{w}) = 0.5$, we may design a loss function which forces $\tilde{m}(\boldsymbol{w})$ to be 0.5 (see Section 4). *Consequently, with a sufficient number of well-spaced counterfactuals to cover $\partial\mathbb{M}$, we may achieve arbitrarily small $|\tilde{m}(\boldsymbol{x}) - m(\boldsymbol{x})|$ at the decision boundary of $m$ (Figure 5).*

Note that, the above mechanism merely aligns the decision boundaries of the target and surrogate models. Hence, to guarantee that the classes are not switched (i.e., to guarantee $m(\boldsymbol{x}) > 0.5 \implies \tilde{m}(\boldsymbol{x}) > 0.5$ and vice versa), the surrogate model needs to be trained on at least some points that lie further away from the decision boundary, in addition to the counterfactuals.

**Remark 3** (Extensions to Other Contours). *This strategy can be extended to approximate any given contour $\partial\mathbb{M}_k = \{\boldsymbol{x} \in [0, 1]^d : m(\boldsymbol{x}) = k\}$. It can be helpful in cases where the counterfactual generating algorithm provides counterfactuals that lie on $\partial\mathbb{M}_k$ for some $k > 0.5$, e.g., due to constraints imposed by robustness (Upadhyay et al., 2021; Hamman et al., 2023; Black et al., 2022), given that the attacker knows the value of $k$.*

It is worth noting that the above strategy overcomes two challenges beset in existing works; (i) the problem of decision boundary shift (particularly with one-sided counterfactuals) present in the method suggested by Aïvodji et al. (2020) and (ii) the need for counterfactuals from both sides of the decision boundary in the methods of Aïvodji et al. (2020) and Wang et al. (2022).

### 3.3 An analysis of the attack based on the Lipschitz continuity

We now present a complexity result for the strategy outlined in Section 3.2. Even though the attack is valid for a decision boundary of any shape, here, we assume conditions similar to Theorem 2 for analytical tractability.

**Theorem 4.** *Consider a pair of $\gamma$-Lipschitz continuous target and surrogate classifiers, $m(\boldsymbol{x})$ and $\tilde{m}(\boldsymbol{x})(\boldsymbol{x} \in [0,1]^d)$, respectively, with $m(\boldsymbol{x})$ having a convex decision boundary (specifically, the set $\{\boldsymbol{x} \in [0,1]^d : \lfloor m(\boldsymbol{x}) \rceil = 1\}$ is convex). Assume the explanation mechanism provides closest counterfactuals. For any point $\boldsymbol{x}$ on the decision boundary of $m$, $|\tilde{m}(\boldsymbol{x}) - m(\boldsymbol{x})| \leq \epsilon$ can be achieved by $\left\lceil 2d \left( \frac{2\gamma\sqrt{d-1}}{\epsilon} - 1 \right)^{d-1} \right\rceil$ number of queries.*

We provide a constructive proof for Theorem 4 in Appendix A.4. As a proof sketch, consider the following line of arguments. First, a net of points (an $\eta$-cover) is constructed over the $(d-1)-$dimensional faces of the $d$-dimensional hypercube (see Figure 6). Next, from Theorem 1, we observe that the closest counterfactual of a point is equivalent to its projection onto the decision boundary. Therefore, the counterfactuals of the $\eta$-cover over the hypercube form an $\eta$-cover over the decision boundary, due to the contraction of distance between points when projected onto a convex hypersurface (Aleksandrov, 1967, Chapter III Lemma 2). The $\eta$-cover over the decision boundary upper-bounds $||\boldsymbol{x} - \boldsymbol{w}||_2$ for all $\boldsymbol{x}$ on the decision boundary, where $\boldsymbol{w}$ is the counterfactual from the $\eta$-cover which is closest to $\boldsymbol{x}$. Substituting this upper-bound in the right-hand side of the inequality in Theorem 3 along with some algebraic manipulations yields the result.

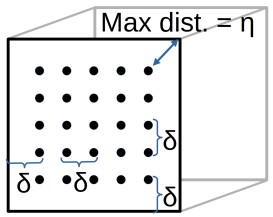

Figure 6: $\eta$-net $\tilde{\mathcal{N}}_\delta$ on a 2-dimensional face of a 3-dimensional unit cube.

Monotonicity of a classifier (see Definition 7) has been identified as a highly desirable property for preserving fairness and explainability (Liu et al., 2020). The classifier being monotonic, in addition to having a convex decision boundary, allows a reduction in the required number of queries, as stated in the corollary below (Appendix A.4 presents a proof).

**Corollary 1.** *Assume $m(\boldsymbol{x})$ to be monotonic in $q(\leq d)$ features, in addition to the assumptions in Theorem 4. Then, for any point $\boldsymbol{x}$ on the decision boundary of $m$, $|\tilde{m}(\boldsymbol{x}) - m(\boldsymbol{x})| \leq \epsilon$ can be achieved by $\left\lceil (2d - q) \left( \frac{2\gamma\sqrt{d-1}}{\epsilon} - 1 \right)^{d-1} \right\rceil$ number of queries.*

## 4 Experiments

While the primary contribution of this work is theoretical, in this section we further present empirical evidence corresponding to the results presented in Section 3.

**Verifying Theorem 2**: We carry out the attack described in Section 3.1 in a synthetic setting where the model has a spherical decision boundary since they are known to be more difficult for polytope approximation (Arya et al., 2012). Figure 7 presents a log-log plot comparing the theoretical and empirical query complexities for several dimensionality values $d$. The empirical approximation error decays faster than $n^{-2/(d-1)}$ as predicted by Theorem 2 (see Appendix B.1 for more details).

Next we present details of the experiments corresponding to the attack proposed in Section 3.2.

**Model architectures:** The classifiers are neural networks whose decision boundaries are not necessarily convex. All the target models use binary cross entropy as the loss function. The surrogate models use either a binary cross entropy

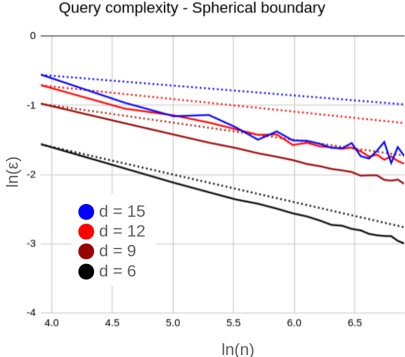

Figure 7: A synthetic experiment for verifying Theorem 2. Dotted and solid lines indicate the theoretical and empirical rates of convergence.

loss (based on Aïvodji et al. (2020), named "Baseline") or a modified version of the binary cross entropy loss which treats counterfactuals differently (named "Proposed"). The modified loss function is given in Equation 5. Let $y(\boldsymbol{x})$ denote the target model's prediction received from the API, associated with the input instance $\boldsymbol{x}$. We assume that the counterfactuals are distinguishable from the ordinary instances, and assign them a label $y(\boldsymbol{w}) = 0.5$. Then,

$$
L_k(\tilde{m}, y) = \frac{1}{|\mathbb{D}|} \sum_{\boldsymbol{x} \in \mathbb{D}} \left( \mathbb{1}\left[y(\boldsymbol{x}) = 0.5, \tilde{m}(\boldsymbol{x}) \leq k\right] \left\{ k \log\left(\frac{k}{\tilde{m}(\boldsymbol{x})}\right) + (1-k) \log\left(\frac{1-k}{1-\tilde{m}(\boldsymbol{x})}\right) \right\} \right.
$$
$$
\left. - \mathbb{1}\left[y(\boldsymbol{x}) \neq 0.5\right] \left\{ y(\boldsymbol{x}) \log\left(\tilde{m}(\boldsymbol{x})\right) + (1 - y(\boldsymbol{x})) \log\left(1 - \tilde{m}(\boldsymbol{x})\right) \right\} \right). \quad (5)
$$

Here, $\mathbb{D}$ denotes the set of all query instances and the counterfactuals. This loss function ensures $\tilde{m}(\boldsymbol{w}) \approx k$ for the counterfactuals $\boldsymbol{w}$. The first term accounts for the counterfactuals, where they are assigned a non-zero loss if the surrogate model's prediction is below $k$. Furthermore, this term averts the effects of counterfactuals that lie farther inside the favorable region, which are the result of imperfections in the generating mechanisms. The second term is the ordinary binary cross-entropy loss, which becomes non-zero only for ordinary query instances. Note that substituting $k = 1$ in $L_k(\tilde{m}, y)$ yields the ordinary binary cross entropy loss. It is noteworthy that this approach is different from the broad area of soft-label learning Nguyen et al. (2011a;b) in two major aspects: (i) the labels in our problem do not smoothly span the interval [0,1] – instead they are either 0, 1 or 0.5; (ii) labels of counterfactuals do not indicate a class probability – the class is strictly $y = 1$, and hence, the counterfactuals that are well within the surrogate decision boundary do not cause a penalty even if their label is $k \neq 1$.

**Experimental setup:** We experiment with two counterfactual generating methods, namely, Diverse Counterfactual Explanations (DiCE) due to Mothilal et al. (2020), and the minimum cost counterfactuals (denoted by MCCF) by Wachter et al. (2017). Fidelity is used for evaluating the agreement between the models. It is evaluated over both uniformly generated instances (denoted by $\mathbb{D}_{\text{uni}}$) and test data instances from the data manifold (denoted by $\mathbb{D}_{\text{test}}$) as the reference dataset $\mathbb{D}_{\text{ref}}$. We compare the "Proposed" surrogate models against the "Baseline" surrogate models.

The experiments were carried out as follows:

1. Randomly initialize the target model and train using $\mathbb{D}_{\text{train}}$.
2. Initialize $\mathbb{A}$ as the placeholder for results from the queries. The results include labels predicted by the target model for all the query instances (i.e., $\{\lfloor m(\boldsymbol{x}) \rceil : \boldsymbol{x} \in \mathbb{D}\}$), and counterfactuals for the query instances with label 0 (i.e., $\{g_m(\boldsymbol{x}) : \boldsymbol{x} \in \mathbb{D}, \lfloor m(\boldsymbol{x}) \rceil = 0\}$).
3. Generate a random seed $R$ for initializing surrogate models.
4. For $t = 1, 2, \ldots, T$ :
   (a) Query for $N$ data points from the attack dataset $\mathbb{D}_{\text{attack}}$ and append the results to $\mathbb{A}$.
   (b) Initialize all surrogate models with $R$ and train on $\mathbb{A}$, where the loss function is $L_1(\tilde{m}, y)$ for "Baseline" models and $L_k(\tilde{m}, y), 0 < k < 1$ for "Proposed" models.
   (c) Record $\ell$ and the corresponding fidelities over $\mathbb{D}_{\text{ref}}$.
5. Repeat steps 1, 2, 3 and 4 for $S$ number of times and calculate average fidelities for each $\ell$, across repetitions.

Based on the experiments of Aïvodji et al. (2020) and Wang et al. (2022), we select $T, N = 20$ and $S = 100$. The value of $k$ was determined empirically. In practice, this can be done based on probability predictions received through the API (Tramèr et al., 2016) or by brute-forcing based on cross-validation scores. For more details on the experimental setup, see Appendix B.

**Results:** Figure 8 illustrates the results of the experiments conducted on a synthetic dataset. In the figure, it is clearly visible that the "Baseline" model is affected by a decision boundary shift. In contrast, the "Proposed" model's decision boundary closely approximates the target decision boundary. More details corresponding to this experiment are given in Appendix B.1.

Table 1 provides the fidelity of "Proposed" and "Baseline" surrogate models on real-world data, evaluated over $\mathbb{D}_{\text{test}}$ and $\mathbb{D}_{\text{uni}}$ as $\mathbb{D}_{\text{ref}}$. In all cases, the "Proposed" surrogate model performs either better or similar to the "Baseline" surrogate model. Additional experiments on real-world datasets are detailed in Appendix B.2.

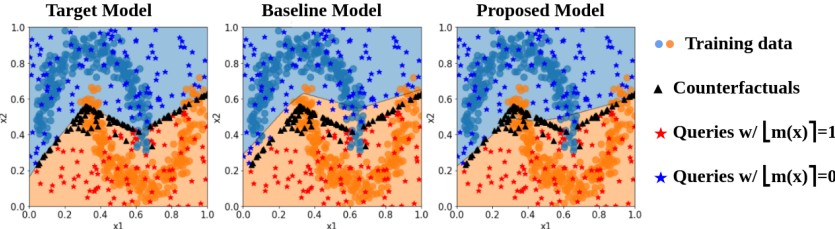

Figure 8: A 2-dimensional demonstration of the proposed attack. Orange and blue shades denote the favorable and unfavorable decision regions of each model. Circles denote the target model's training data. The value of $k$ is set to $k = 0.5$.

Table 1: Fidelity for real-world datasets achieved with 400 queries. Values presented are the averages $\pm$ standard deviations (as percentages) over an ensemble of size 100.

| | | Architecture known (model 0) | | | | Architecture unknown (model 1) | | | |
| --- | --- | --- | --- | --- | --- | --- | --- | --- | --- |
| | | $\mathbb{D}_{test}$ | | $\mathbb{D}_{uni}$ | | $\mathbb{D}_{test}$ | | $\mathbb{D}_{uni}$ | |
| | | Base. | Prop. | Base. | Prop. | Base. | Prop. | Base. | Prop. |
| Adult In. | MCCF | 91$\pm$3.2 | 94$\pm$3.2 | 84$\pm$3.2 | 91$\pm$3.2 | 91$\pm$4.5 | 94$\pm$3.2 | 84$\pm$3.2 | 90$\pm$3.2 |
| | DiCE | 91$\pm$3.2 | 93$\pm$3.2 | 87$\pm$4.5 | 90$\pm$3.2 | 91$\pm$3.2 | 92$\pm$3.2 | 87$\pm$4.5 | 89$\pm$4.5 |
| COMPAS | MCCF | 92$\pm$3.2 | 96$\pm$2.0 | 94$\pm$1.7 | 96$\pm$2.0 | 91$\pm$8.9 | 96$\pm$3.2 | 94$\pm$2.0 | 94$\pm$8.9 |
| | DiCE | 96$\pm$1.7 | 96$\pm$1.7 | 93$\pm$2.0 | 95$\pm$1.7 | 95$\pm$1.7 | 95$\pm$5.5 | 93$\pm$1.7 | 94$\pm$1.7 |
| DCCC | MCCF | 89$\pm$8.9 | 99$\pm$0.9 | 95$\pm$2.2 | 96$\pm$1.4 | 90$\pm$7.7 | 97$\pm$4.5 | 95$\pm$2.2 | 95$\pm$11.8 |
| | DiCE | 98$\pm$1.7 | 98$\pm$1.0 | 94$\pm$1.7 | 95$\pm$1.4 | 97$\pm$2.0 | 98$\pm$1.7 | 94$\pm$1.7 | 95$\pm$1.7 |
| HELOC | MCCF | 91$\pm$4.7 | 96$\pm$2.2 | 92$\pm$2.8 | 94$\pm$2.4 | 90$\pm$7.4 | 95$\pm$5.5 | 91$\pm$3.3 | 93$\pm$3.2 |
| | DiCE | 95$\pm$2.6 | 95$\pm$2.6 | 94$\pm$2.4 | 95$\pm$2.0 | 95$\pm$4.5 | 95$\pm$4.5 | 93$\pm$2.6 | 94$\pm$2.4 |

***Conclusion:*** We introduce two model extraction strategies aided by counterfactuals, along with the analyses of the corresponding query complexities under a few constraints. The two attacks exploit (i) properties of closest counterfactuals; and (ii) the Lipschitz continuity of target and surrogate models, respectively. Our work addresses an important knowledge gap in the existing literature by providing theoretical guarantees. Furthermore, our attacks address the issue of decision boundary shift amidst a system providing one-sided counterfactuals. Experiments demonstrate a significant improvement in fidelity compared to the baseline method proposed in Aïvodji et al. (2020) for the case of one-sided counterfactuals. In a broader notion, we demonstrate that one-sided counterfactuals can be used for perfecting model extraction attacks, exposing a potential vulnerability in MLaaS platforms. Given the importance of counterfactuals in explaining model predictions, we hope our work will inspire countermeasures and defense strategies, paving the way toward secure and trustworthy machine learning systems.

***Limitations and future work:*** One limitation of our theoretical results is the assumption of a convex decision boundary. While this assumption is satisfied in some settings Amos et al. (2017), it is not often the case. In addition, the derivations require the counterfactual generating mechanisms to provide the closest counterfactuals. Any relaxation to these assumptions are paths for exploration. We note that the attack proposed in Section 3.2 remains valid as long as the majority of counterfactuals lie closer to the decision boundary; they need not be the closest ones to the corresponding original instances. Utilizing techniques in active learning in conjunction with counterfactuals is another problem of interest. Extending the results of this work for multi-class classification scenarios can also be explored. Our findings also highlight an interesting connection between Lipschitz constant and vulnerability to model extraction, which could also have implications for future work on generalization, adversarial robustness, etc.

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

## A    PROOFS

### A.1    PROOF OF LEMMA 1 AND THEOREM 1

**Lemma 1.** *Let $\mathcal{S}(\boldsymbol{x}) = 0$ and $\mathcal{T}(\boldsymbol{x}) = 0$ denote two differentiable hypersurfaces in $\mathbb{R}^d$, touching each other at point $\boldsymbol{w}$. Then, $\mathcal{S}(\boldsymbol{x}) = 0$ and $\mathcal{T}(\boldsymbol{x}) = 0$ have a common tangent hyperplane at $\boldsymbol{w}$.*

*Proof.* From Definition 5, there exists a non-empty neighborhood $\mathcal{B}_{\boldsymbol{w}}$ around $\boldsymbol{w}$, such that $\forall \boldsymbol{x} \in \mathcal{B}_{\boldsymbol{w}}$ with $\mathcal{S}(\boldsymbol{x}) = 0$ and $\boldsymbol{x} \neq \boldsymbol{w}$, only one of $\mathcal{T}(\boldsymbol{x}) > 0$ or $\mathcal{T}(\boldsymbol{x}) < 0$ holds. Let $\boldsymbol{x} = (x_1, x_2, \ldots, x_d)$ and $\boldsymbol{x}_{[p]}$ denote $\boldsymbol{x}$ without $x_p$ for $1 \leq p \leq d$. Then, within the neighborhood $\mathcal{B}_{\boldsymbol{w}}$, we may re-parameterize $\mathcal{S}(\boldsymbol{x}) = 0$ as $x_p = S(\boldsymbol{x}_{[p]})$. Note a similar re-parameterization denoted by $x_p = T(\boldsymbol{x}_{[p]})$ can be applied to $\mathcal{T}(\boldsymbol{x}) = 0$ as well. Let $\mathcal{A}_{\boldsymbol{w}} = \{\boldsymbol{x}_{[p]} : \boldsymbol{x} \in \mathcal{B}_{\boldsymbol{w}} \setminus \{\boldsymbol{w}\}\}$. From Definition 5, all $\boldsymbol{x} \in \mathcal{B}_{\boldsymbol{w}} \setminus \{\boldsymbol{w}\}$ satisfy only one of $\mathcal{T}(\boldsymbol{x}) < 0$ or $\mathcal{T}(\boldsymbol{x}) > 0$, and hence without loss of generality the re-parameterization of $\mathcal{T}(\boldsymbol{x}) = 0$ can be such that $S(\boldsymbol{x}_{[p]}) < T(\boldsymbol{x}_{[p]})$ holds for all $\boldsymbol{x}_{[p]} \in \mathcal{A}_{\boldsymbol{w}}$. Now, define $F(\boldsymbol{x}_{[p]}) \equiv T(\boldsymbol{x}_{[p]}) - S(\boldsymbol{x}_{[p]})$. Observe that $F(\boldsymbol{x}_{[p]})$ has a minimum at $\boldsymbol{w}$ and hence, $\nabla_{\boldsymbol{x}_{[p]}} F(\boldsymbol{w}_{[p]}) = 0$. Consequently, $\nabla_{\boldsymbol{x}_{[p]}} T(\boldsymbol{w}_{[p]}) = \nabla_{\boldsymbol{x}_{[p]}} S(\boldsymbol{w}_{[p]})$, which implies that the tangent hyperplanes to both hypersurfaces have the same gradient at $\boldsymbol{w}$. Proof concludes by observing that since both tangent hyperplanes go through $\boldsymbol{w}$, the two hypersurfaces should share a common tangent hyperplane at $\boldsymbol{w}$. ☐

**Theorem 1.** *Let $\mathcal{S}$ denote the decision boundary of a classifier and $\boldsymbol{x} \in [0, 1]^d$ be any point that is not on $\mathcal{S}$. Then, the line joining $\boldsymbol{x}$ and its closest counterfactual $\boldsymbol{w}$ is perpendicular to $\mathcal{S}$ at $\boldsymbol{w}$.*

*Proof.* The proof utilizes the following lemma.

**Lemma 2.** *Consider the $d$-dimensional ball $\mathcal{C}_{\boldsymbol{x}}$ centered at $\boldsymbol{x}$, with $\boldsymbol{w}$ lying on its boundary (hence $\mathcal{C}_{\boldsymbol{x}}$ intersects $\mathcal{S}$ at $\boldsymbol{w}$). Then, $\mathcal{S}$ lies completely outside $\mathcal{C}_{\boldsymbol{x}}$.*

The proof of Lemma 2 follows from the following contradiction. Assume a part of $\mathcal{S}$ lies within $\mathcal{C}_{\boldsymbol{x}}$. Then, points on the intersection of $\mathcal{S}$ and the interior of $\mathcal{C}_{\boldsymbol{x}}$ are closer to $\boldsymbol{x}$ than $\boldsymbol{w}$. Hence, $\boldsymbol{w}$ can no longer be the closest point to $\boldsymbol{x}$, on $\mathcal{S}$.

From Lemma 2, $\mathcal{C}_{\boldsymbol{x}}$ is touching the curve $\mathcal{S}$ at $\boldsymbol{w}$, and hence, they share the same tangent hyperplane at $\boldsymbol{w}$ by Lemma 1. Now, observing that the line joining $\boldsymbol{w}$ and $\boldsymbol{x}$, being a radius of $\mathcal{C}_{\boldsymbol{x}}$, is the normal to the ball at $\boldsymbol{w}$ concludes the proof (see Figure 9). ☐

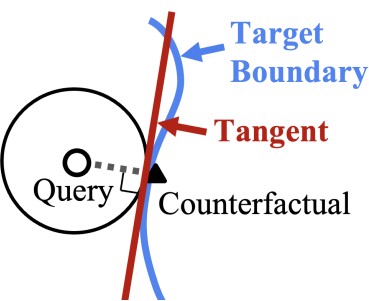

Figure 9: Line joining the query and its closest counterfactual is perpendicular to the decision boundary at the counterfactual. See Theorem 1 for details.

We present the following corollary as an additional observation resulting from Lemma 2.

**Corollary 2.** *Following Lemma 2, it can be seen that all the points in the $d$-dimensional ball with $\boldsymbol{x}$ as the center and $\boldsymbol{w}$ on boundary lies on the same side of $\mathcal{S}$ as $\boldsymbol{x}$.*

## A.2 PROOF OF THEOREM 2

**Theorem 2.** *Let $m$ be the target binary classifier whose decision boundary is convex (i.e., the set $\{\boldsymbol{x} \in [0,1]^d : \lfloor m(\boldsymbol{x}) \rceil = 1\}$ is convex) and has a continuous second derivative. Denote by $\tilde{M}_n$, the polytope approximation of $m$ constructed with $n$ supporting hyperplanes obtained through i.i.d. queries. Assume that the fidelity is evaluated with respect to a $\mathbb{D}_{ref}$ which is uniformly distributed over $[0,1]^d$. Then, when $n \to \infty$ the expected fidelity of $\tilde{M}_n$ is given by*

$$\mathbb{E}\left[\mathrm{Fid}_{m,\mathbb{D}_{ref}}(\tilde{M}_n)\right] = 1 - \epsilon \tag{3}$$

*where $\epsilon \sim \mathcal{O}\left(n^{-\frac{2}{d-1}}\right)$ and the expectation is over both $\tilde{M}_n$ and $\mathbb{D}_{ref}$.*

*Proof.* We first have a look at Böröczky Jr & Reitzner (2004, Theorem 1 (restated as Theorem 5 below)) from the polytope theory. Let $\mathbb{M}$ be a compact convex set with a second-order differentiable boundary denoted by $\partial \mathbb{M}$. Let $\boldsymbol{a}_1, \dots, \boldsymbol{a}_n$ be $n$ randomly chosen points on $\partial \mathbb{M}$, distributed independently and identically according to a given density $d_{\partial \mathbb{M}}$. Denote by $H_+(\boldsymbol{a}_i)$ the supporting hyperplane of $\partial \mathbb{M}$ at $\boldsymbol{a}_i$. Assume $C$ to be a large enough hypercube which contains $\mathbb{M}$ in its interior.

Now, define

$$\tilde{\mathbb{M}}_n = \bigcap_{i=1}^{n} H_+(\boldsymbol{a}_i) \cap C \tag{6}$$

which is the polytope created by the intersection of all the supporting hyperplanes. The theorem characterizes the expected difference of the volumes of $\mathbb{M}$ and $\tilde{\mathbb{M}}_n$.

**Theorem 5** (Random Polytope Approximation, (Böröczky Jr & Reitzner, 2004)). *For a convex compact set $\mathbb{M}$ with second-order differentiable $\partial \mathbb{M}$ and non-zero continuous density $d_{\partial \mathbb{M}}$,*

$$\mathbb{E}\left[V(\tilde{\mathbb{M}}_n) - V(\mathbb{M})\right] = \tau\left(\partial \mathbb{M}, d\right) n^{-\frac{2}{d-1}} + o\left(n^{-\frac{2}{d-1}}\right) \tag{7}$$

*as $n \to \infty$, where $V(\cdot)$ denotes the volume (i.e., the Lebesgue measure), and $\tau(\partial \mathbb{M}, d)$ is a constant that depends only on the boundary $\partial \mathbb{M}$ and the dimensionality $d$ of the space.*

Let $\mathbf{x}_i, i = 1, \dots, n$ be $n$ i.i.d queries from the $\lfloor m(\mathbf{x}) \rceil = 0$ region of the target model. Then, their corresponding counterfactuals $g_m(\mathbf{x}_i)$ are also i.i.d. Furthermore, they lie on the decision boundary of $m$. Hence, we may arrive at the following result.

**Corollary 3.** *Let $\mathbb{M} = \{\boldsymbol{x} \in [0,1]^d : \lfloor m(\boldsymbol{x}) \rceil = 1\}$ and $\tilde{\mathbb{M}}_n = \{\boldsymbol{x} \in [0,1]^d : \lfloor \tilde{M}_n(\boldsymbol{x}) \rceil = 1\}$. Then, by Theorem 5,*

$$\mathbb{E}\left[V(\tilde{\mathbb{M}}_n) - V(\mathbb{M})\right] \sim \mathcal{O}\left(n^{-\frac{2}{d-1}}\right) \tag{8}$$

*when $n \to \infty$. Note that $\mathbb{M} \subseteq \tilde{\mathbb{M}}_n$ and hence, the left-hand side is always non-negative.*

From Definition 3, we may write

$$\mathbb{E}\left[\mathrm{Fid}_{m,\mathbb{D}_{\mathrm{ref}}}(\tilde{M}_n)\right]$$

$$= \mathbb{E}\left[\frac{1}{|\mathbb{D}_{\mathrm{ref}}|} \sum_{\mathbf{x} \in \mathbb{D}_{\mathrm{ref}}} \mathbb{E}\left[\mathbb{1}\left[\lfloor m(\mathbf{x}) \rceil = \left\lfloor \tilde{M}_n(\mathbf{x}) \right\rceil\right] \Big| \mathbb{D}_{\mathrm{ref}}\right]\right] \tag{9}$$

$$= \frac{1}{|\mathbb{D}_{\mathrm{ref}}|} \mathbb{E}\left[\sum_{\mathbf{x} \in \mathbb{D}_{\mathrm{ref}}} \mathbb{P}\left[\lfloor m(\mathbf{x}) \rceil = \left\lfloor \tilde{M}_n(\mathbf{x}) \right\rceil \Big| \mathbf{x}\right]\right] \quad (\because \text{ query size is fixed}) \tag{10}$$

$$= \mathbb{P}\left[\lfloor m(\mathbf{x}) \rceil = \left\lfloor \tilde{M}_n(\mathbf{x}) \right\rceil\right] \quad (\because \mathbf{x}\text{'s are i.i.d.}) \tag{11}$$

$$= \int_{\mathcal{M}_n} \mathbb{P}\left[\lfloor m(\mathbf{x}) \rceil = \left\lfloor \tilde{M}_n(\mathbf{x}) \right\rceil \Big| \tilde{M}_n(\mathbf{x}) = \tilde{m}_n(\mathbf{x})\right] \mathbb{P}\left[\tilde{M}_n(\mathbf{x}) = \tilde{m}_n(\mathbf{x})\right] \mathrm{d}\tilde{m}_n \tag{12}$$

where $\mathcal{M}_n$ is the set of all possible $\tilde{m}_n$'s.

Now, by noting that

$$\mathbb{P}\left[\lfloor m(\mathbf{x})\rceil = \left\lfloor \tilde{M}_n(\mathbf{x})\right\rceil \middle| \tilde{M}_n(\mathbf{x}) = \tilde{m}_n(\mathbf{x})\right] = 1 - \mathbb{P}\left[\lfloor m(\mathbf{x})\rceil \neq \left\lfloor \tilde{M}_n(\mathbf{x})\right\rceil \middle| \tilde{M}_n(\mathbf{x}) = \tilde{m}_n(\mathbf{x})\right],$$
(13)

we may obtain

$$\mathbb{E}\left[\mathrm{Fid}_{m,\mathbb{D}_{\mathrm{ref}}}(\tilde{M}_n)\right] = 1 - \int_{\mathcal{M}_n} \mathbb{P}\left[\lfloor m(\mathbf{x})\rceil \neq \left\lfloor \tilde{M}_n(\mathbf{x})\right\rceil \middle| \tilde{M}_n(\mathbf{x}) = \tilde{m}_n(\mathbf{x})\right]$$
$$\times \mathbb{P}\left[\tilde{M}_n(\mathbf{x}) = \tilde{m}_n(\mathbf{x})\right] \mathrm{d}\tilde{m}_n \quad (14)$$

$$= 1 - \int_{\mathcal{M}_n} \underbrace{\frac{V(\tilde{\mathbb{M}}_n) - V(\mathbb{M})}{\text{Total volume}}}_{=1 \text{ for unit hypercube}} \mathbb{P}\left[\tilde{M}_n(\mathbf{x}) = \tilde{m}_n(\mathbf{x})\right] \mathrm{d}\tilde{m}_n$$

$$(\because \mathbf{x}\text{'s are uniformly distributed}) \quad (15)$$

$$= 1 - \mathbb{E}\left[V(\tilde{\mathbb{M}}_n) - V(\mathbb{M})\right]. \quad (16)$$

The above result, in conjunction with Corollary 3, concludes the proof. $\square$

## A.3 Proof of Theorem 3

**Theorem 3.** *Suppose the target ($m(\boldsymbol{x})$) and the surrogate ($\tilde{m}(\boldsymbol{x})$) models are $\gamma$-Lipschitz continuous. Assume $m(\boldsymbol{w}) = \tilde{m}(\boldsymbol{w})$ for some $\boldsymbol{w} \in [0,1]^d$. Then, for any $\boldsymbol{x} \in [0,1]^d$, the difference between the outputs of the two models is bounded from above as follows;*

$$|\tilde{m}(\boldsymbol{x}) - m(\boldsymbol{x})| \leq 2\gamma||\boldsymbol{x} - \boldsymbol{w}||_2. \quad (4)$$

*Proof.*

$$|\tilde{m}(\boldsymbol{x}) - m(\boldsymbol{x})| = |\tilde{m}(\boldsymbol{x}) - \tilde{m}(\boldsymbol{w}) - (m(\boldsymbol{x}) - \tilde{m}(\boldsymbol{w}))| \quad (17)$$
$$= |\tilde{m}(\boldsymbol{x}) - \tilde{m}(\boldsymbol{w}) - (m(\boldsymbol{x}) - m(\boldsymbol{w}))| \quad (18)$$
$$\leq \underbrace{|\tilde{m}(\boldsymbol{x}) - \tilde{m}(\boldsymbol{w})|}_{\leq \gamma||\boldsymbol{x}-\boldsymbol{w}||_2} + \underbrace{|m(\boldsymbol{x}) - m(\boldsymbol{w})|}_{\leq \gamma||\boldsymbol{x}-\boldsymbol{w}||_2} \quad (19)$$
$$\leq 2\gamma||\boldsymbol{x} - \boldsymbol{w}||_2 \quad (20)$$

where the first inequality is a result of applying the triangle inequality and the second follows from the definition of Lipschitz continuity (Definition 6). $\square$

## A.4 Proof of Theorem 4 and Corollary 1

**Theorem 4.** *Consider a pair of $\gamma$-Lipschitz continuous target and surrogate classifiers, $m(\boldsymbol{x})$ and $\tilde{m}(\boldsymbol{x})(\boldsymbol{x} \in [0,1]^d)$, respectively, with $m(\boldsymbol{x})$ having a convex decision boundary (specifically, the set $\{\boldsymbol{x} \in [0,1]^d : \lfloor m(\boldsymbol{x})\rceil = 1\}$ is convex). Assume the explanation mechanism provides closest counterfactuals. For any point $\boldsymbol{x}$ on the decision boundary of $m$, $|\tilde{m}(\boldsymbol{x}) - m(\boldsymbol{x})| \leq \epsilon$ can be achieved by $\left\lceil 2d\left(\frac{2\gamma\sqrt{d-1}}{\epsilon} - 1\right)^{d-1}\right\rceil$ number of queries.*

*Proof.* An $\eta$-covering over the faces of the unit hypercube can be constructed as follows. Consider a net of points, $\tilde{\mathcal{N}}_\delta$, on a given face of the unit hypercube, such that the outermost points are $\delta$ away from the $(d-2)$-dimensional edges and each point is $\delta$ away from its closest neighbors (see Figure 6). The cardinality of this net can be calculated as follows;

$$|\tilde{\mathcal{N}}_\delta| = \# \text{ points on a face} \quad (21)$$
$$= (\# \text{ points along a single dimension of a face})^{d-1} \quad (22)$$
$$= \left(\frac{\text{length of a side}}{\text{gap between points}} - 1\right)^{d-1} \quad (23)$$
$$= \left(\frac{1}{\delta} - 1\right)^{d-1}. \quad (24)$$

Note that any point on the $(d-1)$-dimensional face of the hypercube is no further than $\sqrt{d-1}\delta$ from a point belonging to $\tilde{\mathcal{N}}_\delta$. Consequently, if we have similar nets on each face of the $d$-dimensional unit hypercube, the maximum distance from any point on the surface of the hypercube to a point belonging to the composite net $\mathcal{N}_\delta$ is $\sqrt{d-1}\delta$. Since the hypercube has $2d$ faces, $|\mathcal{N}_\delta| = 2d|\tilde{\mathcal{N}}_\delta| = 2d\left(\frac{1}{\delta} - 1\right)^{d-1}$. Letting $\eta = \sqrt{d-1}\delta$ and simplifying for $|\mathcal{N}_\delta|$ gives

$$\mathcal{N}_\delta = \left\lceil 2d\left(\frac{\sqrt{d-1}}{\eta} - 1\right)^{d-1}\right\rceil. \tag{25}$$

Now, by querying explanations for the points belonging to this $\eta$-cover on the hypercube, we obtain an $\eta$-cover over the convex decision boundary of the target model. This is a consequence of combining Theorem 1 with the fact that projecting the points on the faces of the hypercube onto the convex hypersurface would only reduce the distance between them (Aleksandrov, 1967, Chapter III Lemma 2). The $\eta$-cover over the decision boundary guarantees $||\boldsymbol{x} - \boldsymbol{w}||_2 \leq \eta$, where $\boldsymbol{x}$ is any point on the target decision boundary and $\boldsymbol{w}$ is its closest counterfactual. We may obtain the final result by letting $\eta = \epsilon/2\gamma$ and by replacing the right-hand side of Theorem 3 with the new upper-bound. $\qquad\square$

**Corollary 1.** *Assume $m(\boldsymbol{x})$ to be monotonic in $q(\leq d)$ features, in addition to the assumptions in Theorem 4. Then, for any point $\boldsymbol{x}$ on the decision boundary of $m$, $|\tilde{m}(\boldsymbol{x}) - m(\boldsymbol{x})| \leq \epsilon$ can be achieved by $\left\lceil (2d-q)\left(\frac{2\gamma\sqrt{d-1}}{\epsilon} - 1\right)^{d-1}\right\rceil$ number of queries.*

*Proof.* Assume that $m(\boldsymbol{x})$ is monotonic in the feature $x_j$ (see Definition 7). Then, from the two faces of the unit hypercube at $x_j = 0$ and $x_j = 1$, it is sufficient to query only from the face at $x_j = 0$, since the query points on the other face will result counterfactuals that lie on the face itself. Note that the information provided by such counterfactuals is already available by knowing that $x_j$ is monotonic. Extending the argument to all the $q$ monotonic features will reduce queries from $q$ faces, which concludes the proof. $\qquad\square$

## B  EXPERIMENTAL DETAILS AND ADDITIONAL RESULTS

In this section, we provide details about our experimental setup with additional experimental results.

### B.1  DETAILS OF SYNTHETIC EXPERIMENTS

We carry-out two synthetic experiments: one to demonstrate the validity of Theorem 2 and the other to visualize the attack proposed in Section 3.2.

**Experiment for verifying Theorem 2:** This experiment includes approximating a spherical decision boundary in the first quadrant of a $d-$dimensional space. The decision boundary is a portion of a sphere with radius 1 and the origin at $(1, 1, \ldots, 1)$. The input space is assumed to be normalized, and hence, restricted to the unit hypercube. See Section 3.1 for a description of the attack strategy. Figure 10 presents a visualization of the experiment in the case where $d = 2$. Figure 7 presents a comparison of theoretical and empirical query complexities for $d$ greater than 2.

**Experiment to visualize attack proposed in Section 3.2:** This experiment is conducted on a synthetic dataset which consists of 1000 samples generated using the `make_moons` function from the `sklearn` package. Features are normalized to the range $[0, 1]$ before feeding to the classifier. Model architectures are given below:

Target model:

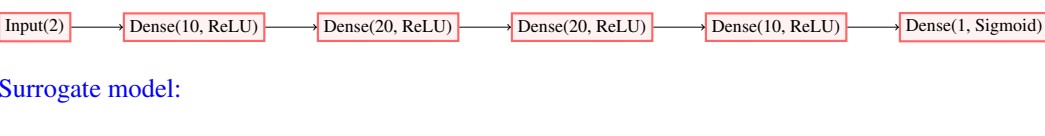

Surrogate model:

Input(2) → Dense(10, ReLU) → Dense(20, ReLU) → Dense(20, ReLU) → Dense(1, Sigmoid)

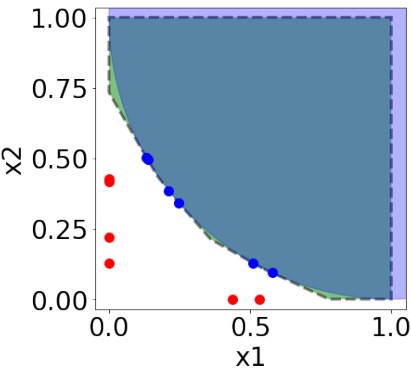

Figure 10: Synthetic attack for verifying Theorem 2 in the 2-dimensional case. Red dots represent queries and blue dots are the corresponding closest counterfactuals. Dashed lines indicate the boundary of the polytope approximation.

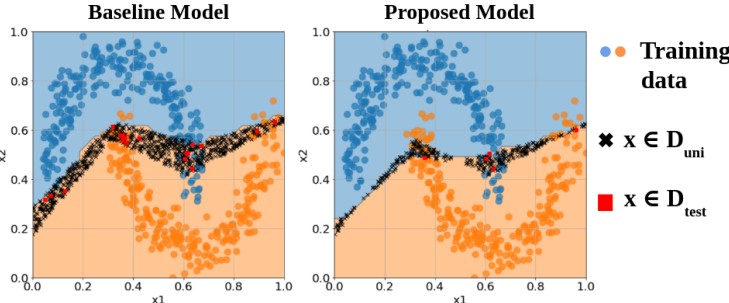

Figure 11: Misclassifications w.r.t. to the target model, over $\mathbb{D}_{uni}$ and $\mathbb{D}_{test}$ as the reference datasets for the 2-dimensional demonstration in Figure 8. "Baseline" model causes a large number of misclassifications w.r.t. the "Proposed" model.

All the layers are L2-regularized with a regularization coefficient of 0.001. Each model is trained for 100 epochs with a batch size of 32. Since the intention of this experiment is to demonstrate the functionality of the modified loss function given in equation 5, a large query of size 200 is used, instead of performing multiple small queries. An empirically determined value of $k = 0.5$ was used, along with MCCF as the counterfactual generating method. Figure 8 shows how the original model extraction attack proposed by Aïvodji et al. (2020) suffers from the boundary shift issue, while the model with the proposed loss function overcomes this problem. Figure 11 illustrates the instances misclassified by the two surrogate models.

## B.2 Details of Experiments on Real-World Datasets

We use four publicly available real-world tabular datasets (namely, Adult Income, COMPAS, DCCC, and HELOC) to evaluate the performance of the proposed attack. The details of these datasets are discussed next.

- Adult Income: The dataset is a 1994 census database with information such as educational level, marital status, age and annual income of individuals (Becker & Kohavi, 1996). The target is to predict "income", which indicates whether the annual income of a given person exceeds \$50000 or not (i.e., $y = \mathbb{1}[\text{income} \geq 0.5]$). It contains 32561 instances in total (the training set), comprising of 24720 from $y = 0$ and 7841 from $y = 1$. To make the dataset class-wise balanced we randomly sample 7841 instances from class $y = 0$, giving a total effective size of 15682 instances. Each instance has 6 numerical features and 8 categorical features. During pre-processing, categorical features are encoded as integers. All the features are then normalized to the range $[0, 1]$.

- Home Equity Line of Credit (HELOC): This dataset contains information about customers who have requested a credit line as a percentage of home equity FICO (2018). It contains 10459 instances with 23 numerical features each. Prediction target is "is_at_risk" which indicates whether a given customer would pay the loan in the future. Dataset is slightly unbalanced with class sizes of 5000 and 5459 for $y = 0$ and $y = 1$, respectively. Instead of using all 23 features, we use the following subset of 10 for our experiments; "estimate_of_risk", "net_fraction_of_revolving_burden", "percentage_of_legal_trades", "months_since_last_inquiry_not_recent", "months_since_last_trade", "percentage_trades_with_balance", "number_of_satisfactory_trades", "average_duration_of_resolution", "nr_total_trades", "nr_banks_with_high_ratio". ALl the features are normalized to lie in the range $[0, 1]$.

- Correctional Offender Management Profiling for Alternative Sanctions (COMPAS): This dataset has been used for investigating racial biases in a commercial algorithm used for evaluating reoffending risks of criminal defendants (Angwin et al., 2016). It includes 6172 instances and 20 numerical features. The target variable is "is_recid". Class-wise counts are 3182 and 2990 for $y = 0$ and $y = 1$, respectively. All the features are normalized to the interval $[0, 1]$ during pre-processing.

- Default of Credit Card Clients (DCCC): The dataset includes information about credit card clients in Taiwan Yeh (2016). The target is to predict whether a client will default on the credit or not, indicated by "default.payment.next.month". The dataset contains 30000 instances with 24 attributes each. Class-wise counts are 23364 from $y = 0$ and 6636 from $y = 1$. To alleviate the imbalance, we randomly select 6636 instances from $y = 0$ class, instead of using all the instances. All the attributes are numerical, and normalized to $[0, 1]$ during pre-processing.

Two surrogate model architectures, one exactly similar to the target architecture (model 0 - known architecture) and the other slightly different (model 1 - unknown architecture), are tested. Model architectures are as follows:

Target model/Surrogate model 0:

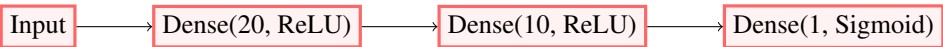

Surrogate model 1:

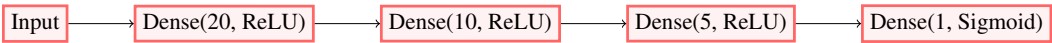

Each layer is L2-regularized with a regularization coefficient of 0.001. Each model is trained for 200 epochs with a batch size of 32. Values of $k$ used were 0.5 for MCCF and 0.8 for DiCE.

**Primary results:** Figures 12 and 14 illustrate the fidelities achieved by the two model architectures described above. Figures 13 and 15 show the corresponding variances of the fidelity values over 100 realizations. It can be observed that the variances diminish as the query size grows, indicating more stable model extractions. Figure 16 compares the rate of convergence of the empirical approximation error i.e., $1 - \mathbb{E}\left[\text{Fid}_{m,\mathbb{D}_{\text{ref}}}(\tilde{M}_n)\right]$ with the rate predicted by Theorem 5. Notice how the empirical error decays faster than $n^{-2/(d-1)}$.

Figure 17 illustrates the dependence of the attack performance on the Lipschitz constant of the target model. Following Gouk et al. (2021), we approximate the Lipschitz constant of a neural network by the product of the spectral norms of the weight matrices. We achieve different Lipschitz constants by controlling the L2-regularization of the layer weights while training the target model. The plots indicate that a higher Lipschitz constant affects adversely for model extraction.

**Additional results:** We perform additional experiments on a variety of model architectures and counterfactual generating mechanisms. These experiments demonstrate the generalizability of the attack proposed in Section 3.2. Experimental details are provided in Table 2 while Figure 18 illustrated the results. In all the cases, the "Proposed" models surpass "Baseline" models in performance.

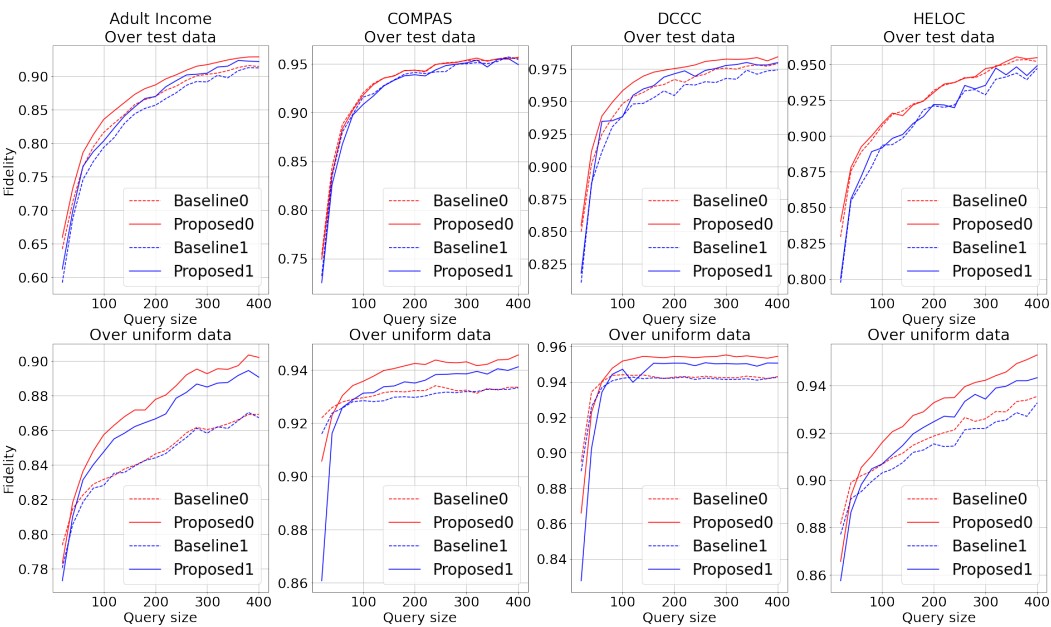

Figure 12: Fidelity for real-world datasets with DiCE as the counterfactual generating mechanism. Solid lines indicate "Proposed" models. Dashed lines indicate "Baseline" models. Colors correspond to the model architecture.

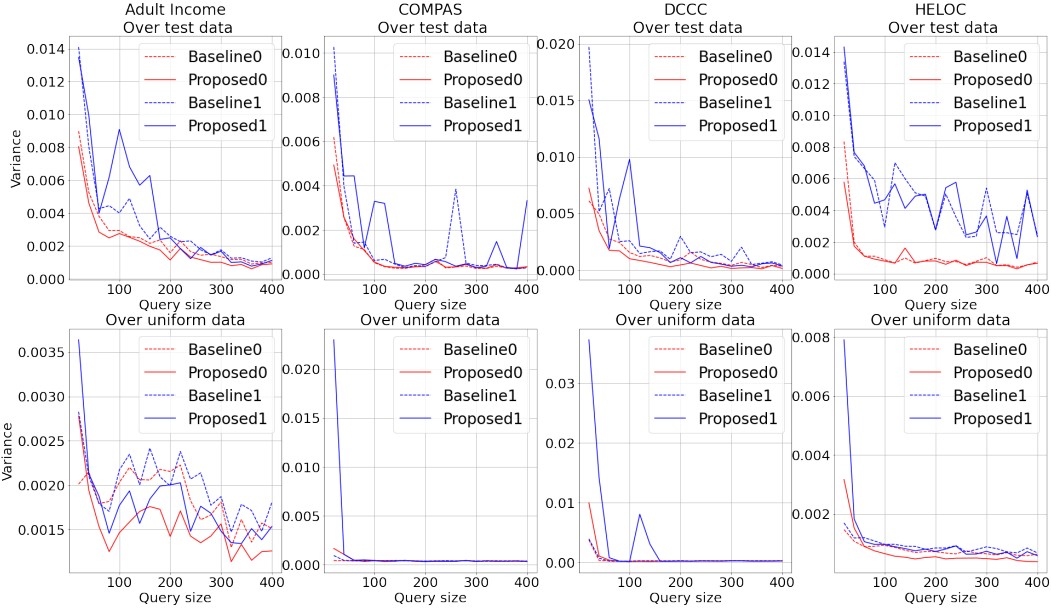

Figure 13: Variance of fidelity for real-world datasets with DiCE as the counterfactual generating mechanism. Solid lines indicate "Proposed" models. Dashed lines indicate "Baseline" models. Colors correspond to the model architecture.

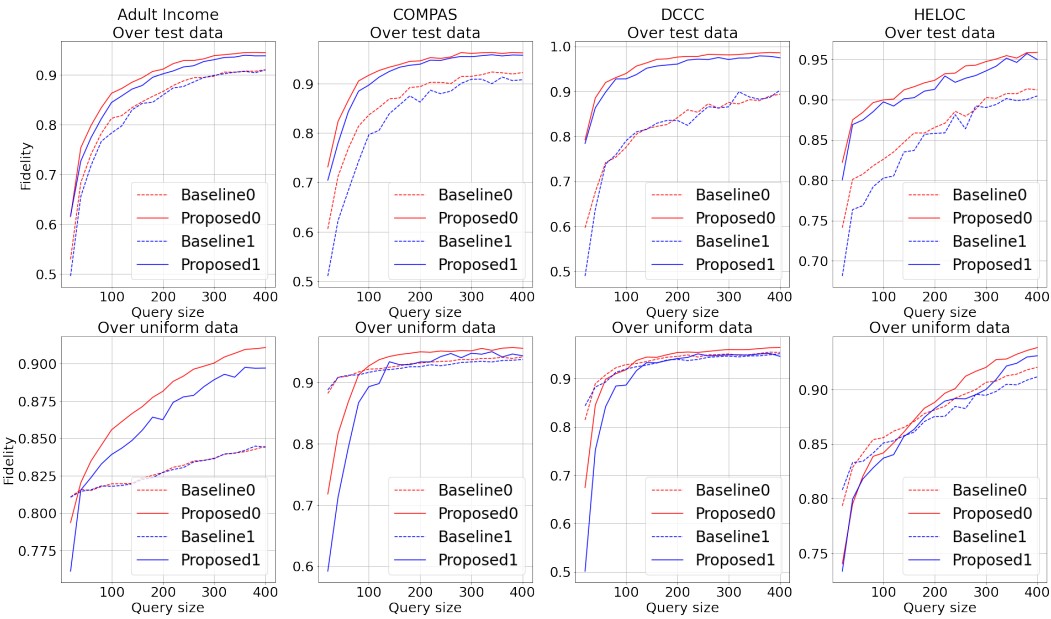

Figure 14: Fidelity for real-world datasets with MCCF as the counterfactual generating mechanism. Solid lines indicate "Proposed" models. Dashed lines indicate "Baseline" models. Colors correspond to the model architecture.

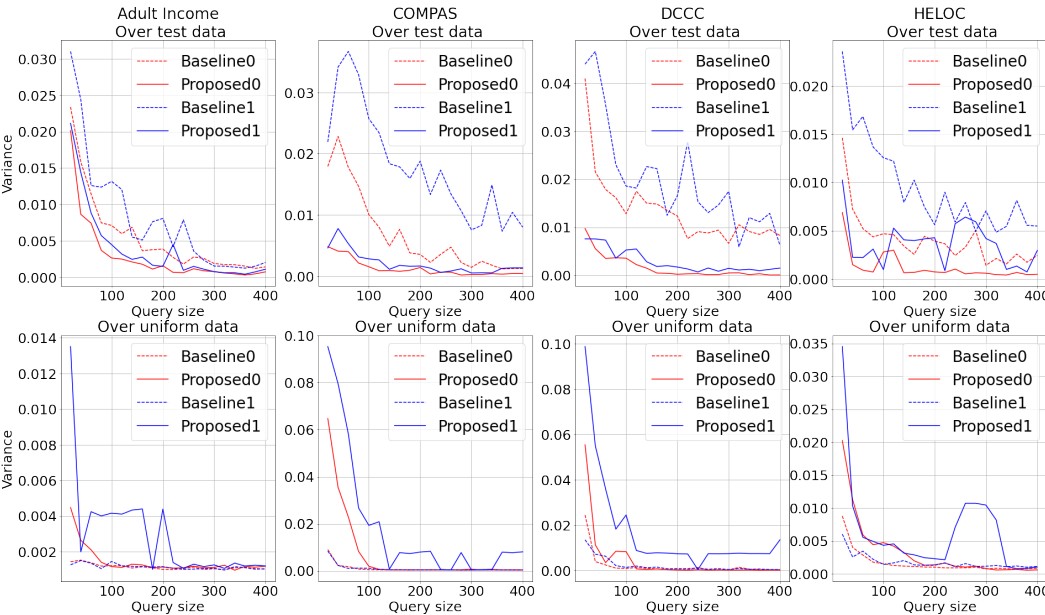

Figure 15: Variance of fidelity for real-world datasets with MCCF as the counterfactual generating mechanism. Solid lines indicate "Proposed" models. Dashed lines indicate "Baseline" models. Colors correspond to the model architecture.

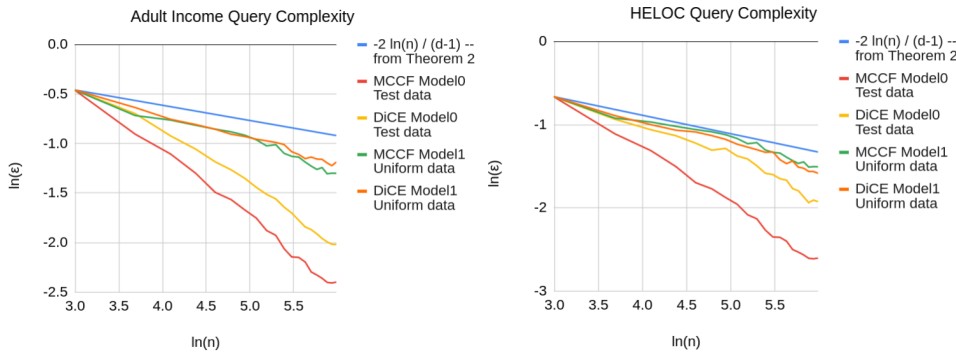

Figure 16: A comparison of the query complexity derived in Theorem 2 with the empirical query complexities obtained on the Adult Income and HELOC datasets. The graphs are on a log-log scale. We observe that the analytical query complexity is an upper bound for the empirical query complexities. All the graphs are recentered with an additive constant for presentational convenience. However, this does not affect the slope of the graph, which corresponds to the complexity.

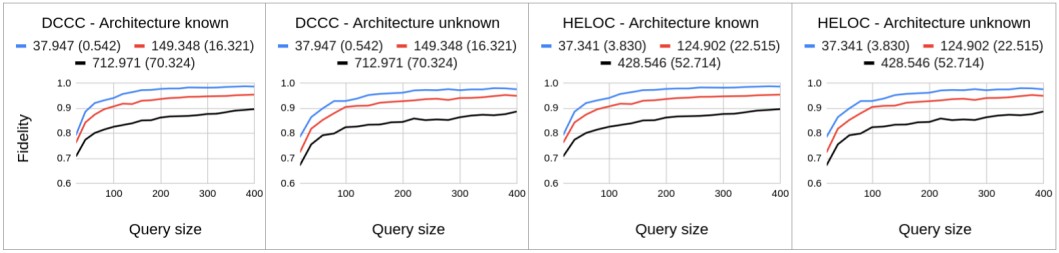

Figure 17: Dependence of fidelity on the target model's Lipschitz constant. The approximations of the Lipschitz constants are shown in the legend with standard deviations within brackets. Lipschitz constants are approximated as the product of the spectral norm of weight matrices in each model. MCCF is used as the counterfactual generating method. With a higher Lipschitz constant, the fidelity achieved by a given number of queries tend to degrade.

Table 2: Details of additional experiments demonstrating the generalizability of the proposed attack. See Figure 18 for results.

|  | Dataset | Counterfactual generating method | Hidden layer sizes |
|---|---|---|---|
| Experiment A | COMPAS | MCCF L2-norm | Target: 10, 30, 10 
 Surrogate0: 10, 10 
 Surrogate1: 10, 20 |
| Experiment B | DCCC | MCCF L1-norm | Target: 20, 30, 10 
 Surrogate0: 30, 30, 10 
 Surrogate1: 10, 20 
 Surrogate2: 20, 30, 10 |
| Experiment C | Adult Income | MCCF L1-norm | Target: 10, 30, 10 
 Surrogate0: 10 
 Surrogate1: 10, 30, 20 |
| All the hidden layers are dense and the activations are ReLU. L2 regularization with a coefficient of 0.001 has been used. All the output layer activations are Sigmoid. | | | |

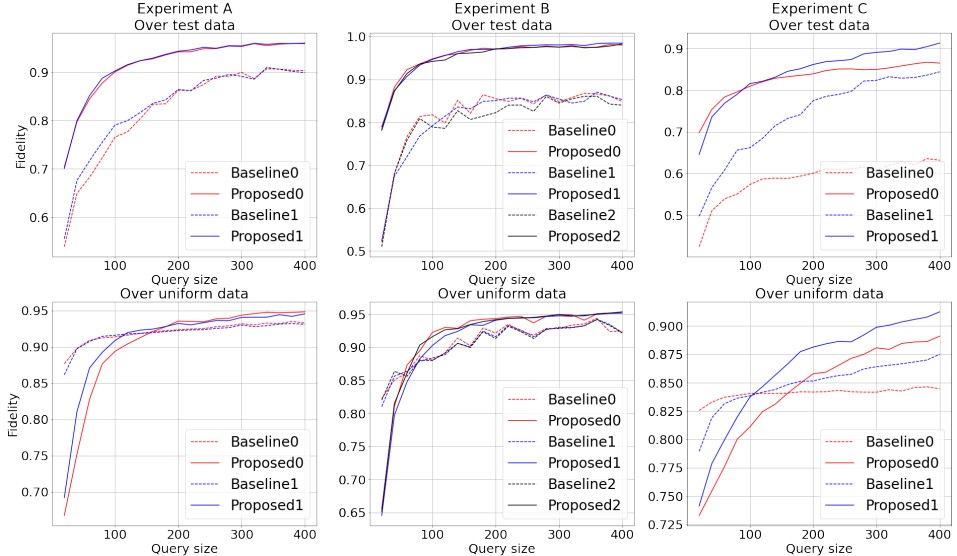

Figure 18: Additional experiments on different model architectures and counterfactual generating methods. Details are given in Table 2. Results indicate that the performance is independent of model architecture as long as the surrogate models are closer to the target model in complexity, which is also an observation made by Aïvodji et al. (2020). Furthermore, the "Proposed" models perform better than the "Baseline" models irrespective of the type of the cost function $c(\boldsymbol{x}, \boldsymbol{w})$ used in the counterfactual generation mechanism, given that the generated counterfactuals lie closer to the decision boundary.

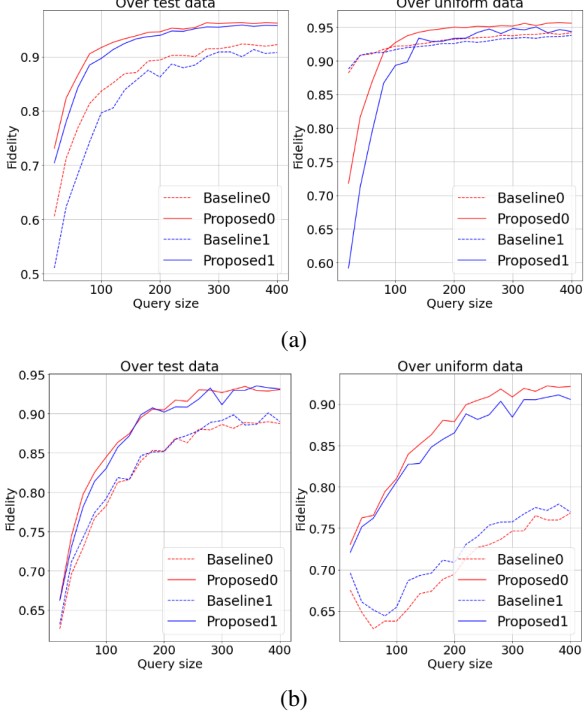

Figure 19: Results corresponding to the Adult Income dataset with queries sampled from biased versions of the dataset (i.e., a biased $\mathbb{D}_{\text{attack}}$). The version used in Figure 19a contains 24720 and 7841 examples from classes $y = 0$ and $y = 1$, respectively. The version corresponding to Figure 19b contains 2472 and 7841 examples from classes $y = 0$ and $y = 1$, respectively.

