# OpenReview forum: "The Role of Counterfactual Explanations in Model Extraction Attacks"
_ICLR.cc/2024/Conference — Submitted to ICLR 2024_

### Official Review · Reviewer_4YsF · 2023-10-25

**Soundness:** 3 good
**Presentation:** 3 good
**Contribution:** 2 fair
**Rating:** 5
**Confidence:** 4

**Summary:**

The paper studies how counterfactual explanations can be used for model extraction attacks.

**Strengths:**

- Highly relevant research question
- Paper is well structure and mostly well written (only a discussion of limitation of the proposed method is missing)
- Nice intuitive geometric approach, although I think that there are some limitations (see Section "Weaknesses")

**Weaknesses:**

- Only closest counterfactuals are considered. However, in practice, plausibility and actionability are also very important aspects of recourse. Some explanation generation method might not output closest counterfactuals but plausible ones.
- Lipschitz continuity and monotonicity are both very strong assumptions -- they might not hold for many models in practice.
- Authors should also discuss limitations of their proposed method

**Questions:**

- How can the proposed method be extended to deal with plausible and actionable counterfactuals? As mentioned above, these two aspects are highly relevant in recourse, and might be used instead of closest counterfactuals.
- What to do in case of really large Lipschitz constants? Would the proposed method still work?

---

> ### Author Response · Authors · 2023-11-17
> **Response to Reviewer 4YsF (1/2)**
>
> We appreciate the recognition of the relevance of our work. Based on the comments, we make the following edits to our paper.
>
> **On other counterfactual-generation mechanisms:** We have now included results from a rich set of experiments covering different model architectures and counterfactual generating methods (see Section 4 and Appendix B.2). Particularly, the results cover sparse counterfactuals which are generated by using the L1-norm as the cost function in the counterfactual generating method. Moreover, we would like to point out that the proposed practical attack is independent of the counterfactual generating method, as long as the generated counterfactuals lie sufficiently closer to the decision boundary. They need not be the closest ones to the corresponding original instances. We included this in the limitations section (see page 9).
>
> **On the assumptions of Lipschitz continuity and monotonicity:** We note that the practical neural networks may have large global Lipschitz constants. However, we first make the observation that our theoretical guarantee (Theorem 3) would also extend to the local Lipschitz continuity, which may be more common as compared to global Lipschitz continuity (added Remark 2 on page 6). In fact, this observation leads to another interesting interpretation of our theoretical results that is generally insightful even for practical neural networks: *the regions of the decision boundary with low local-Lipschitz constant are more easily reconstructed.*
>
> Furthermore, we have now included a new set of experimental results demonstrating the relationship  of the attack performance with the Lipschitz constant (see Figure 17 in Appendix B.2 page 21). Following [1], we approximate the Lipschitz constant of a neural network by the product of the spectral norms of the weight matrices. We achieve different Lipschitz constants by controlling the L2-regularization of the layer weights while training the target model. Our approximations were in the range 37 to 713 and are also comparable to the values present in other works [1, Figure 4],[2, Figure 1],[3, Table 1]. Therefore, it is evident from the experiments that the proposed attack remains valid for significantly large Lipschitz constants. Our observations follow our theoretical insights, where a greater Lipschitz constant gradually degrades the fidelity for a given number of queries.
>
> Our findings are interesting and significant because they highlight an interesting connection between Lipschitz constant and vulnerability to model extraction. Furthermore, noting that a low Lipschitz constant is an indicator of good generalization of a model and adversarial robustness [1],[2],[3], our findings also point towards a potential tradeoff between good generalization and vulnerability to model extraction attacks, which could also be interesting for future research.
>
> Moreover, we note that monotonicity is a favorable characteristic in the context of fair and interpretable machine learning [4], which is achievable particularly in the case of tabular data. We would like to clarify that monotonicity is not assumed in case of the main theorems or in the proposed practical attack or experiments. We only assume monotonicity to show an additional reduction in the query complexity as pointed out in Corollary 1.

---

> ### Author Response · Authors · 2023-11-17
> **Response to Reviewer 4YsF (2/2)**
>
> **On the discussion on limitations:** Following the suggestions of Reviewer 4YsF we have now included this section in our modified manuscript (see page 9).
>
> “Limitations and future work: One limitation of our theoretical results is the assumption of a convex decision boundary. While this assumption is satisfied in some settings [5], it is not often the case. In addition, the derivations require the counterfactual generating mechanisms to provide the closest counterfactuals. Any relaxation to these assumptions are paths for exploration. We note that the attack proposed in Section 3.2 remains valid as long as the majority of counterfactuals lie closer to the decision boundary; they need not be the closest ones to the corresponding original instances. Utilizing techniques in active learning in conjunction with counterfactuals is another problem of interest. Extending the results of this work for multi-class classification scenarios can also be explored. Our findings also highlight an interesting connection between Lipschitz constant and vulnerability to model extraction, which could also have implications for future work on generalization, adversarial robustness, etc.”
>
> [1] H. Gouk, E. Frank, B. Pfahringer, and M. J. Cree, “Regularisation of neural networks by enforcing Lipschitz continuity,” Mach Learn, vol. 110, no. 2, pp. 393–416, Feb. 2021.
>
> [2] M. Jordan and A. G. Dimakis, “Exactly Computing the Local Lipschitz Constant of ReLU Networks,” in Advances in Neural Information Processing Systems, Curran Associates, Inc., 2020, pp. 7344–7353.
>
> [3] P. Pauli, A. Koch, J. Berberich, P. Kohler and F. Allgöwer, "Training Robust Neural Networks Using Lipschitz Bounds," in IEEE Control Systems Letters, vol. 6, pp. 121-126, 2022.
>
> [4] X. Liu, X. Han, N. Zhang, and Q. Liu, “Certified Monotonic Neural Networks,” in Advances in Neural Information Processing Systems, Curran Associates, Inc., 2020, pp. 15427–15438.
>
> [5] B. Amos, L. Xu, and J. Z. Kolter, “Input Convex Neural Networks,” in Proceedings of the 34th International Conference on Machine Learning, PMLR, Jul. 2017, pp. 146–155.

---

### Official Review · Reviewer_q134 · 2023-10-27

**Soundness:** 3 good
**Presentation:** 3 good
**Contribution:** 3 good
**Rating:** 6
**Confidence:** 4

**Summary:**

This paper presents a theoretical analysis and practical approach to model extraction using counterfactual explanations.
In the theoretical analysis, the authors investigate the query complexity of model extraction under ideal assumptions, where the positive region is convex, and the model is Lipschitz.
In terms of practical methods, the authors focus on the observation that many counterfactual explanations are only required in the direction how the negative decisions can be altered to positive ones.
The authors proposed a method for model extraction from such biased counterfactual explanations.
Specifically, they assumed that counterfactual explanations lie on a decision boundary where the prediction probability is constant, and proposed a modified binary cross-entropy loss based on this assumption.
In the experiments, the authors demonstrated that the proposed method outperformed conventional methods, particularly in addressing the phenomenon known as "decision boundary shift."

**Strengths:**

This paper exhibits two key strengths: a theoretical analysis and a development of practical method for model extraction using counterfactual explanations.
In theoretical analysis, the authors established ideal assumptions such as a convex positive region and a Lipschitz model. Even under these ideal conditions, it would be important to theoretically elucidate query complexity, leading to an important step forward in the advancement of model extraction research.
Regarding the practical method, the authors demonstrated that "decision boundary shift" can be easily addressed by a simple modification of the cross-entropy loss function.

**Originality, Quality**

This study demonstrates originality in identifying conditions such as a convex positive region and a Lipschitz model as factors influencing query complexity in model extraction.
While these conditions may be limited and idealized, the theoretical clarification of query complexity remains a crucial step in moving the model extraction research forward.

**Clarity**

Throughout the paper, the fundamental concepts and contributions of the research are clearly stated.

**Significance**

Demonstrating that "decision boundary shift" can be addressed through a straightforward modification of the cross-entropy loss is considered particularly significant.
On the other hand, the theoretical analysis of query complexity is still confined to ideal conditions, necessitating further in-depth analysis for this result to have a broad impact within the research field.

**Weaknesses:**

This paper has two weaknesses: "Novelty and Effectiveness of the Modified Cross-Entropy Loss" and "Gap Between Theoretical Analysis and Methodology."

**Novelty and Effectiveness of the Modified Cross-Entropy Loss**

An inherent aspect of counterfactual explanations is that they correspond to points on the decision boundary, making it natural to consider their class probability as 0.5.
When using counterfactual explanations as training data for model extraction, it is also natural to use them as data points with a class probability of 0.5.
Viewing the problem of model extraction in this light, it can be reduced to a learning problem involving soft labels.
The question that arises is whether the modified cross-entropy loss proposed in this study is a novel and particularly effective method for this problem.
While this is not necessarily a fatal weakness, the paper appears to lack a discussion regarding the relevance of the modified cross-entropy loss to the broader context of soft label research and its unique effectiveness and novelty in this specific problem.

**Gap Between Theoretical Analysis and Methodology**

In the theoretical analysis, the assumptions of the convexity of the positive region and the Lipschitz property of the model play essential roles.
However, these assumptions are entirely disregarded in Section 4.
While it is understood that models in real world do not exhibit convex positive regions, it appears that the paper currently contains two largely independent sections: one focused on theory and the other on methodology.
In Section 4, the authors mention, "While the primary contribution of this work is theoretical, in this section we further present an empirical performance evaluation of the model extraction attack devised in the previous section for one-sided counterfactuals."
This statement likely refers to the content in Section 3.2.
However, Section 4 lacks explicit mention of the Lipschitz property.
Therefore, questions arise about whether the Lipschitz assumption plays a fundamental role in Section 4, as suggested in Section 3.2.

**Questions:**

* Are there any other possible alternatives of the proposed modified cross entropy loss, in particular in the context of soft label? If not, what is the essential difference or advantage of the proposed the proposed modified cross entropy loss?
* Do the Lipschitz assumption plays a fundamental role in Section 4, as suggested in Section 3.2? Or, all we need is the proposed modified cross entropy loss alone?

---

> ### Author Response · Authors · 2023-11-17
> **Response to Reviewer q134**
>
> We are grateful to Reviewer q134 for their review and for pointing out our key strengths, i.e., “a theoretical analysis and a development of practical method for model extraction using counterfactual explanations”. As they have clearly identified, we provide novel results in a setting where the counterfactual explanations are provided for queries from only one class.
>
> **On the novelty and effectiveness of the proposed loss function:** We note that the problem setting of the broadly studied area of soft-label learning [1],[2] is different from the problem we have at hand due to the following two main reasons:
>
> (I) The labels in our problem do not smoothly span the interval [0,1]. Instead they are either 0, 1 or 0.5.
>
> (II) The label of 0.5 that is assigned to counterfactuals does not necessarily indicate a class probability. The class of a counterfactual is strictly y = 1 as it is a training example for the surrogate model. The counterfactuals that are well within the surrogate decision boundary should not cause a penalty even if their label is 0.5.
>
> Based on the suggestion of Reviewer q134, we have now included the above clarification in the manuscript under “Model architectures” in Section 4 (page 7).
>
> In fact, in our early experiments we have tried using a loss function that symmetrically penalizes counterfactual explanations if $\tilde{m}(\boldsymbol{w}) \neq 0.5$. This, however, did not provide expected results mainly due to some counterfactuals being  well within the boundary caused by imperfections in the generating method. The proposed loss function (Equation (5) page 8) alleviates this issue by not penalizing such counterfactual explanations.
>
> **On the gap between theoretical analysis and methodology:** We have addressed the comments in two ways. First, we present a synthetic experiment validating the sample complexity derived in Theorem 2 (see Figure 7 in Section 4 and Appendix B.1 page 16). For our experiment, we specifically choose a spherical decision boundary which is known to be particularly difficult for polytope approximation [3]. Our experiments demonstrate that for each choice of dimension $d$, the expected approximation error $\epsilon =1-\mathbb{E}[\text{Fidelity}]$ decays faster than $n^{-2/(d-1)}$ where $n$ is the query size. We demonstrate this using a log-log plot of ln($\epsilon$) vs ln($n$) (see Figure 7 page 7).
>
> Next, we present the results of a new set of experiments based on real-world data, which demonstrates the effect of the Lipschitz constant of the target model on the attack performance (see Figure 17 Appendix B.2 page 21). Following [4], we approximate the Lipschitz constant of a neural network by the product of the spectral norms of the weight matrices. We achieve different Lipschitz constants by controlling the L2-regularization of the layer weights while training the target model. Our approximations were in the range 37 to 713 and are also comparable to the values obtained in other works [4, Figure 4],[5, Figure 1],[6, Table 1]. Therefore, it is evident from the experiments that the proposed attack remains valid for significantly large Lipschitz constants. Our observations follow our theoretical insights, where a greater Lipschitz constant gradually degrades the fidelity for a given number of queries.
>
> [1] Q. Nguyen, H. Valizadegan, A. Seybert, and M. Hauskrecht, “Sample-efficient learning with auxiliary class-label information,” AMIA Annu Symp Proc, vol. 2011, pp. 1004–1012, 2011.
>
> [2] Q. Nguyen, H. Valizadegan, and M. Hauskrecht, “Learning classification with auxiliary probabilistic information,” Proc IEEE Int Conf Data Min, vol. 2011, pp. 477–486, 2011.
>
> [3] S. Arya, G. D. Da Fonseca, and D. M. Mount, “Polytope Approximation and the Mahler Volume,” in Proceedings of the Twenty-Third Annual ACM-SIAM Symposium on Discrete Algorithms, Society for Industrial and Applied Mathematics, Jan. 2012, pp. 29–42.
>
> [4] H. Gouk, E. Frank, B. Pfahringer, and M. J. Cree, “Regularisation of neural networks by enforcing Lipschitz continuity,” Mach Learn, vol. 110, no. 2, pp. 393–416, Feb. 2021.
>
> [5] M. Jordan and A. G. Dimakis, “Exactly Computing the Local Lipschitz Constant of ReLU Networks,” in Advances in Neural Information Processing Systems, Curran Associates, Inc., 2020, pp. 7344–7353.
>
> [6] P. Pauli, A. Koch, J. Berberich, P. Kohler and F. Allgöwer, "Training Robust Neural Networks Using Lipschitz Bounds," in IEEE Control Systems Letters, vol. 6, pp. 121-126, 2022.

---

> > ### Comment · Reviewer_q134 · 2023-11-23
> > **Re: Response to Reviewer q134**
> >
> > I would like to thank the authors for detailed comments and additional experiments.
> >
> > Here, I would like to add one comment on soft-labels.
> > In Eq.(5), the authors considered a customized loss function.
> > I wonder whether this can be simplified as the standard cross entropy loss.
> > $$
> > L(m, y) = - \sum_{(x, y) \in D} \left(y \log m(x) + (1 - y) \log (1 - m(x)) \right)
> > $$
> > In the current problem, as the authors noted, there are three possible $y$: $y = 0, 1$, and $0.5$.
> > For $y = 0.5$, the cross entropy loss $- \left(y \log m(x) + (1 - y) \log (1 - m(x)) \right)$ is minimized when $m(x) = 0.5$.
> > So, I think just by simply learning the surrogate $m$ by the standard cross entropy loss will keep the boundary point with $y = 0.5$ as the boundary on the surrogate as $m(x) = 0.5$.

---

> ### Author Response · Authors · 2023-11-23
> **Re: Re: Response to Reviewer q134**
>
> We thank Reviewer q134 for their review and for their positive opinion about our work.
>
> **Regarding the cross-entropy loss for counterfactuals:**
> In our early experiments, we experimented with the standard binary cross entropy loss for counterfactuals with label 0.5. This loss function is symmetric around 0.5. However, we found that this symmetric loss function did not provide good model approximation results. Instead, our proposed loss function only penalizes when $\tilde{m}(x) < 0.5$ for counterfactuals but not if $\tilde{m}(x) \geq 0.5$. We have found that this asymmetric loss function achieves good model approximation as demonstrated in our experimental results. We believe this happens because practical counterfactual generation mechanisms generate counterfactuals that are close to the boundary while being on the accepted side $(m(x) \geq 0.5)$ but may not be exactly 0.5.
>
> We include some examples from our experiments here (the dataset is Adult Income and the counterfactual generating method is DiCE, fidelities are for surrogate model 0):
>
> Query size: 100 ----> Ordinary BCE loss: 0.424 -----> Proposed loss: 0.835
>
> Query size: 200 ----> Ordinary BCE loss: 0.519 -----> Proposed loss: 0.887
>
> Query size: 300 ----> Ordinary BCE loss: 0.532 -----> Proposed loss: 0.917

---

### Official Review · Reviewer_MQ33 · 2023-10-30

**Soundness:** 2 fair
**Presentation:** 2 fair
**Contribution:** 2 fair
**Rating:** 3
**Confidence:** 4

**Summary:**

This paper introduces two model extraction strategies that use counterfactuals. The first strategy exploits the properties of closest counterfactuals, while the second strategy leverages the Lipschitz continuity of target and surrogate models. The authors provide theoretical guarantees and address the issue of decision boundary shift in a system that provides one-sided counterfactuals. Experimental results show improved fidelity compared to a baseline method.

**Strengths:**

1. Remark 1 gives a false sense about the difficulty of the problem. The difficulty of approximating a non-convex body is way more difficult than this. A convex body can be approximated by the intersection of many half spaces, and this is what the authors are using in Section 3.1. However, it is unclear how to approximate a non-convex body using half spaces without logical rules (like boolean variables). Figure 4 is two-dimensional, and we can easily identify the intersections of two hyperplanes; however, for higher dimensions, it is absolutely not clear how to find these intersections. Thus, a similar representation using red piecewise linear curve like Figure 4 does not generalize to higher dimensions.
There is no clear message in Section 3.2: Theorem 3 is too simple to be a theorem, and there is no definitive description of the extraction strategy.
Section 3: the authors claim ``Even though the attack is valid for a decision boundary of any shape”, but there is no clear direction on how to generalize the attack to a non-convex boundary.
The extraction attack omits the queries that are predicted 0 by $m$ in the objective function (5). Why?
The paper provides some sample complexity results for the extraction attacks.

**Weaknesses:**

1. Remark 1 gives a false sense of the difficulty of the problem. The difficulty of approximating a non-convex body is way more difficult than this. A convex body can be approximated by the intersection of many half-spaces, and this is what the authors are using in Section 3.1. However, it is unclear how to approximate a non-convex body using half spaces without logical rules (like boolean variables). Figure 4 is two-dimensional, and we can easily identify the intersections of two hyperplanes; however, for higher dimensions, it is absolutely not clear how to find these intersections (the intersection is again a $d-2$ dimensional manifold). Thus, a similar representation using red piecewise linear curve like Figure 4 does not generalize to higher dimensions.
2. There is no clear message in Section 3.2: Theorem 3 is too simple to be a theorem, and there is no definitive description of the extraction strategy.
3. Section 3: the authors claim ``Even though the attack is valid for a decision boundary of any shape”, but there is no clear direction on how to generalize the attack to a non-convex boundary.
4. The extraction attack omits the queries that are predicted 0 by $m$ in the objective function (5). Why?

Overall, I could not find a strong argument to justify why this paper should be accepted at ICLR. The theoretical contribution of this paper is quite limited, and somewhat trivial by combining existing results in the literature. The practical contribution of this paper is also not strong.

**Questions:**

1. Concerning Remark 2: suppose that MLaaS is using a robust recourse method. How can we estimate the value of $k$ so that we can perform model extraction?
2. Figure 7 and 8 are too simple. Could the authors provide some illustrations for (i) harder dataset, and (ii) $\tilde m$ of lower model complexity than $m$?
3. How would the model behave for an unbalanced dataset, or when the number of counterfactuals is small?
4. Is there any easy active learning method to query that can help us perform an extraction attack with a limited number of targeted queries?

---

> ### Author Response · Authors · 2023-11-16
> **Response to Reviewer MQ33 (1/2)**
>
> We thank Reviewer MQ33 for taking out the time to review this paper. We have made the changes discussed below, in an effort to address the comments.
>
> **On the clarity of Remark 1:** We would like to clarify that our intention is not to downplay the difficulty of approximating a non-convex decision boundary but to simply highlight that concave regions are also amenable to polytope approximation, and we have updated Remark 1 accordingly to reflect this.
>
> "Remark 1: Even though Theorem 2 assumes convexity of the decision boundary for analytical tractability, the attack can be extended to a concave decision boundary. This is because the closest counterfactual will always lead to a tangent hyperplane irrespective of convexity and now the rejected region can be seen as the intersection of these half-spaces (Theorem 1 does not assume convexity). However, it is worth noting that approximating a concave decision boundary is, in general, more difficult than approximating a convex region. To obtain equally-spaced out tangent hyperplanes on the decision boundary, a concave region will require a much denser set of query points (see Figure 4) due to the inverse effect of length contraction discussed in Aleksandrov (1967, Chapter III Lemma 2). Furthermore, approximating a decision boundary which is neither convex nor concave is much more challenging as the decision regions can no longer be approximated as intersections of half-spaces. This motivates us to propose an attack that does not depend on convexity assumption leveraging Lipschitz continuity, as discussed next. Experiments indicate that the query complexity of this Lipschitz-based attack is upper-bounded by the result in Theorem 2 (see Figure 16)."
>
> **On 2D visualizations on more complex datasets:** Following the comments by Reviewer MQ33, we have now replaced Figures 7 and 8 (Figures 8 and 11 in the updated version) with results of an experiment including a more complex dataset ("moons" from sklearn package) and a surrogate model which is simpler than the target model (see Section B.1.)
>
> **On clarifying the attack proposed in Section 3:** We would like to clarify that the practical attack based on the Lipschitz continuity proposed in Section 3.2 does not assume the decision boundary to have any particular shape (convex or concave). It is formulated based on the insights of Theorem 3. However, when analyzing the query complexities for this attack, we assume the decision boundary to be convex. The proposed objective function (Equation (5), also given below) plays a vital role in ensuring the conditions of Theorem 3, where the surrogate model needs to be forced into predicting k ((0,1]) for counterfactual instances.
>
> $L_k(\tilde{m},y) = \frac{1}{|\mathbb{D}|} \sum_{\boldsymbol{x} \in \mathbb{D}} \Bigg( \mathbb{1}[y(\boldsymbol{x})=0.5, \tilde{m}(\boldsymbol{x}) \leq k] \Bigg[k \log\left(\frac{k}{\tilde{m}(\boldsymbol{x})}\right) + (1-k)\log\left(\frac{1-k}{1-\tilde{m}(\boldsymbol{x})}\right) \Bigg]$
> $- \mathbb{1}\left[ y(\boldsymbol{x})\neq 0.5 \right] \Bigg[ y(\boldsymbol{x}) \log\left(\tilde{m}(\boldsymbol{x})\right) + (1-y(\boldsymbol{x})) \log\left(1-\tilde{m}(\boldsymbol{x})\right)\Bigg] \Bigg)$
>
> This is achieved by the first term of the objective function, where all counterfactual explanations whose predictions are less than k are penalized. We do not penalize the counterfactuals that have a prediction larger than k, due to the fact that counterfactual generating methods sometimes generate explanations that are well within the decision boundary of the target model due to imperfections. All the ordinary instances (both 0’s and 1’s) are treated in the same way as a normal classifier would do. In particular, the queries that are predicted 0 by the target model $m(\boldsymbol{x})$ will be treated as ordinary labeled examples.
>
> **On biased datasets and active learning:** We now include results of experiments on biased versions of the Adult Income dataset in Figure 19 in Appendix B.2 page 22. We observe that the proposed attack surpasses the baseline even with the biases present. The proposed attack can be used along with many existing active learning strategies as the attack  does not assume anything on the query generation. We acknowledge that combining active learning methods with exploiting counterfactual explanations for model extraction is an interesting research direction in the limitations section (see page 9).

---

> ### Author Response · Authors · 2023-11-16
> **Response to Reviewer MQ33 (2/2)**
>
> **On Remark 3 (previously Remark 2):** We would like to clarify that Remark 3 (previously Remark 2) is not the main focus of our paper, but a possible extension as we explicitly clarify now. We would only like to point out that the attack can be executed to approximate any contour of constant probability, given that the attacker knows the value for k. This provides some flexibility to mount a model extraction attack on an MLaaS that uses a robust recourse method. In future work, the problem of discovering the value of k in this may be approached in two ways as suggested under “Experimental setup” in Section 4 (page 8): (I) the attacker can tune k based on the cross-validation score achieved on the query set; (II) k can be directly known if the MLaaS provides probabilistic information as considered in [1].
>
> **On the organization of Section 3:** Section 3 of the paper is devoted to three main objectives. First, in Section 3.1, we introduce a novel sample complexity result for a model extraction attack which exploits closest counterfactual explanations. Then, in an effort to relax the conditions in the attack considered under Section 3.1, we state an important observation under Theorem 3 in Section 3.2, along with a novel attack strategy based on this observation. Considering the comments raised by Reviewer MQ33, we have now explicitly indicated this strategy with a topic “Proposed attack”. Finally, in Section 3.3, we analyze the query complexity of this attack under conditions similar to Section 3.1.
>
> [1] F. Tramèr, F. Zhang, A. Juels, M. K. Reiter, and T. Ristenpart, “Stealing Machine Learning Models via Prediction APIs,” presented at the 25th USENIX Security Symposium (USENIX Security 16), 2016, pp. 601–618.

---

### Official Review · Reviewer_vcnE · 2023-10-31

**Soundness:** 2 fair
**Presentation:** 3 good
**Contribution:** 2 fair
**Rating:** 6
**Confidence:** 3

**Summary:**

This paper studies model reconstruction attacks by using the proximity of counterfactuals to the decision boundary. The authors aim to establish theoretical guarantees for such attacks. To this end, they characterize the number of queries required for the attacker to achieve a given error in model approximation using results from polytope theory (Theorem 2). The authors’ main result from Theorem 2 relies on the decision boundary being convex. To relax the convexity assumption, the paper additionally assumes Lipschitz continuity of the underlying model to provide approximation bounds, which depend on the Lipschitz constant which is typically unknown.
Finally, the authors propose a strategy for model extraction.

While the paper offers some strengths in terms of proposing new tools to analyze model extraction attacks, there are several weaknesses that require improvement, including a limited evaluation and theoretical results that are (mostly) confined to models with convex decision boundaries. Overall, the paper provides a good starting point for future research in the area of model extraction attacks through counterfactual explanations but further improvements are necessary to meaningfully generealize the analysis to general non-linear models.

**Strengths:**

- **New theoretical approach to study extraction attacks**: The paper introduces a fresh approach to studying model extraction attacks using counterfactual explanation algorithms, employing methodologies from polytope theory that I have not seen explored in this context before.
-  **New method**: The authors propose a new model extraction method.
- **Clearly structured**: The paper is overall well written and clearly structured.

**Weaknesses:**

**The empirical evaluation is limited**: (1) The paper lacks comparison with more recent model extraction techniques via counterfactual explanations, such as those by Wang et al. (2022). (2) Further, the dependence on dimensionality ($d$) is a critical factor influencing convergence (e.g., see Theorem 2), yet the paper lacks results concerning this aspect in the empirical evaluation of their attack. (3) There is also a disparity between the primary theoretical results (Theorem 2) and the subsequent sections of the paper. (4) I would expect that experimental results verify the validity of Theorem 2. This requires to fit a model with a convex decision boundary and to execute the proposed attack. Finally, (5) the attacks are exclusively conducted using one type of neural network and might not generalize well to other models or other network architectures. For example, it would strenghten the paper's empirical results to explore the suggested method's impact of varying model parameters, etc. on fidelity.

**Confined theoretical results**: (1) The paper does not to adequately reconcile the theoretical analysis with commonly used sparsity-inducing loss functions in standard counterfactual explanation methods. (2) The main (interesting) theoretical results are confined to convex decision boundaries. Hence, the analysis on the number of required queries to reach a given error in fidelity (see Theorem 2) might be substantiialy underestimated.

**Questions:**

- Can the authors provide insights into their attempts to estimate the Lipschitz constant in practical applications? Considering the inherent difficulty in obtaining low Lipschitz constants for neural networks of reasonable size, the practicality of the Lipschitz result may be questionable.
- How did the authors determine the criteria for retaining explanations when generating multiple explanations using the method from Mothilal et al. (2020)?
- Can the authors offer empirical verification of Theorem 2 to strengthen its validity?
- It would be valuable to visualize the convergence rate from Theorem 2 in Figure 9, allowing for an evaluation of whether the empirical predictions for convex boundaries closely align with the empirical behavior for non-convex models.

---

> ### Author Response · Authors · 2023-11-16
> **Response to Reviewer vcnE (1/2)**
>
> We thank Reviewer vcnE for their review and appreciate their acknowledgement of the novelty of our contribution. As Reviewer vcnE points out, our work “introduces a fresh approach to studying model extraction attacks using counterfactual explanation algorithms, employing methodologies from polytope theory that I have not seen explored in this context before.”
>
> **On empirical verification of Theorem 2:** Based on the comment of Reviewer vcnE, we first include the results of a synthetic experiment that empirically validates Theorem 2 across several choices of dimension $d$ (see Section 4 page 7 and Appendix B.1). For our experiment, we specifically choose a spherical decision boundary which is known to be particularly difficult for polytope approximation [1]. Our experiments demonstrate that for each choice of dimension $d$, the expected approximation error $\epsilon = 1-\mathbb{E}[\text{Fidelity}]$ decays faster than $n^{-2/(d-1)}$ where $n$ is the query size. We demonstrate this using a log-log plot of ln($\epsilon$) vs ln($n$) (see Figure 7 in Section 4 and Appendix B.1).
>
> We note that carrying-out the same attack on real data is constrained by the limitation of the counterfactual generation mechanisms that they do not always provide the closest counterfactuals, even if they are configured to do so. We have now discussed this in a limitation section (see page 9):
>
> This limitation leads us to propose an alternative and more practical attack under Section 3.2 that has guarantees under Lipschitz assumptions. Nonetheless, we find that the number of queries obtained using Theorem 2 is not an underestimation, and is in fact higher compared to the actual number of queries required to achieve a certain fidelity for non-convex, real-world datasets using our alternative attack (see Figure 16 in Appendix B.2).
>
> **On the Lipschitz assumption:** We note that the practical neural networks may have large global Lipschitz constants. However, we first make the observation that our theoretical guarantee (Theorem 3) would also extend to the local Lipschitz continuity, which may be more common as compared to global Lipschitz continuity (added Remark 2 on page 6). In fact, this observation leads to another interesting interpretation of our theoretical results that is generally insightful even for practical neural networks: *the regions of the decision boundary with low local-Lipschitz constant are more easily reconstructed.*
>
> Furthermore, we have now included a new set of experimental results demonstrating the relationship of the attack performance with the Lipschitz constant (see Figure 17 in Appendix B.2 page 21). Following [2], we approximate the Lipschitz constant of a neural network by the product of the spectral norms of the weight matrices. We achieve different Lipschitz constants by controlling the L2-regularization of the layer weights while training the target model. Our approximations were in the range 37 to 713 and are also comparable to the values obtained in other works [2, Figure 4],[3, Figure 1],[4, Table 1]. Therefore, it is evident from the experiments that the proposed attack remains valid for significantly large Lipschitz constants. Our observations follow our theoretical insights, where a greater Lipschitz constant gradually degrades the fidelity for a given number of queries.
>
> Our findings are interesting and significant because they highlight an interesting connection between Lipschitz constant and vulnerability to model extraction. Furthermore, noting that a low Lipschitz constant is an indicator of good generalization of a model and adversarial robustness [2],[3],[4], our findings also point towards a potential tradeoff between good generalization and vulnerability to model extraction attacks, which could be interesting for future research.
>
> We further clarify that while our theoretical guarantees are in terms of the Lipschitz constant, we are not required to estimate the Lipschitz constant in our actual implementation of the attack.
>
> **On other architectures and counterfactual-generation mechanisms:** We have now included results from a rich set of experiments covering different model architectures and counterfactual generating methods (see Section 4 and Appendix B.2). Particularly, the results cover sparse counterfactuals (using L1-norm as the cost function in the counterfactual generating method) and a comparison of the convergence rate from Theorem 2 with that corresponding to real-world datasets (see Figure 16 in Appendix B.2). In the case of DiCE counterfactual generating mechanism (by Mothilal et al. [5]), we set the diversity factor to 1 and obtain only one counterfactual explanation per query.

---

> ### Author Response · Authors · 2023-11-16
> **Response to Reviewer vcnE (2/2)**
>
> **On the comparison to Wang et al. (2022):** We had discussed Wang et al. [6] in our related works and have now further emphasized on their contribution in our updated version. Wang et al. [6] is a significant contribution in the area of model extraction for counterfactuals by introducing a clever strategy of further querying for the counterfactual of the counterfactual. *However, this strategy can only be applied when one is allowed to query for counterfactuals from both the accepted and rejected regions.* Our problem setup deviates from this scenario and only allows for one-sided counterfactuals, i.e., counterfactuals only for the rejected points. This is motivated by the fact that a user who has received a favorable decision no longer requires any additional guidance/justification for the decision. Because of this mismatch in the problem setup, we are unable to perform a fair comparison with Wang et al. [6], and only compare with the original model extraction attack of Aїvodji et al. [7] for our one-sided problem setup.
>
> [1] S. Arya, G. D. Da Fonseca, and D. M. Mount, “Polytope Approximation and the Mahler Volume,” in Proceedings of the Twenty-Third Annual ACM-SIAM Symposium on Discrete Algorithms, Society for Industrial and Applied Mathematics, Jan. 2012, pp. 29–42.
>
> [2] H. Gouk, E. Frank, B. Pfahringer, and M. J. Cree, “Regularisation of neural networks by enforcing Lipschitz continuity,” Mach Learn, vol. 110, no. 2, pp. 393–416, Feb. 2021.
>
> [3] M. Jordan and A. G. Dimakis, “Exactly Computing the Local Lipschitz Constant of ReLU Networks,” in Advances in Neural Information Processing Systems, Curran Associates Inc., 2020, pp. 7344–7353.
>
> [4] P. Pauli, A. Koch, J. Berberich, P. Kohler and F. Allgöwer, "Training Robust Neural Networks Using Lipschitz Bounds," in IEEE Control Systems Letters, vol. 6, pp. 121-126, 2022.
>
> [5] R. K. Mothilal, A. Sharma, and C. Tan, “Explaining machine learning classifiers through diverse counterfactual explanations,” in Proceedings of the 2020 Conference on Fairness, Accountability, and Transparency, in FAT* ’20. New York, NY, USA: Association for Computing Machinery, Jan. 2020, pp. 607–617.
>
> [6] Y. Wang, H. Qian, and C. Miao, “DualCF: Efficient Model Extraction Attack from Counterfactual Explanations,” in 2022 ACM Conference on Fairness, Accountability, and Transparency, Seoul Republic of Korea: ACM, Jun. 2022, pp. 1318–1329.
>
> [7] U. Aïvodji, A. Bolot, and S. Gambs, “Model extraction from counterfactual explanations.” arXiv, Sep. 03, 2020.

---

### Author Response · Authors · 2023-11-16

We thank all the reviewers for taking out the time to review our paper. We appreciate that the reviewers have found our contribution to be novel and important.

We analyze model extraction attacks using (one-sided) counterfactual explanations by further leveraging the fact that the counterfactuals are quite close to the decision boundary. Without making any assumptions on the model class, we first examine the fidelity of model extraction through the lens of volume approximation in polytope theory. Accordingly, we provide theoretical guarantees on volume approximation assuming the accepted region is convex, as well as, alternative guarantees under Lipschitz assumptions. An interesting interpretation of our theoretical results is that the regions of the decision boundary with low local-Lipschitz constant are more easily reconstructed. Our findings are significant because they highlight an interesting connection between Lipschitz constant and vulnerability to model extraction, which could also have implications for future work on generalization, adversarial robustness, etc.

Based on the reviewers’ comments, we have updated our paper with several changes, including additional experimental results that provide further insights. Below, we provide a detailed response to each of the reviewers’ comments and the changes made.

---

### Meta-Review · Area_Chair_jPp7 · 2023-12-12

**Metareview:**

This paper studies the viability of model extraction using counterfactual explanations. The authors present a new class of attacks that exploit the fact that counterfactuals must lie near the decision boundary. In this setting, they characterize the number of queries that are required to reconstruct the model under various assumptions on the underlying model (e.g., convex decision boundary, Lipschitz continuity). The paper includes experimental results highlighting the viability of model extraction under these conditions.

*Strengths*

- Clarity: The paper is well-written - with clear figures, notation, and mathematical exposition.
- Topic: This is an important topic with potential implications for privacy and consumer protection.

*Weaknesses*

- Significance: The paper has several weaknesses that broadly affect its significance. The proposed attacks and their analyses hinge on strong assumptions on the model form that are often not met in the classes of applications where we would care about potential reconstruction -- e.g., convex decision boundaries, and Lipschitz continuity would not hold for models like XGBoost, which are widely used in the kinds of applications where we would care about model extraction (e.g., loan approval). In this vein, the experiments would be far stronger if we could see how the proposed attacks perform under a representative set of models, and understand the downstream implications of reconstructions on individuals.

*What's Missing*

The paper would benefit from an overhauled set of experiments that study the viability of model reconstruction under a range of model classes (e.g., including RFs, XGB) and realistic operational considerations (e.g., actionability, individualized cost functions, returning counterfactuals from anchor points). Such experiments would be extremely valuable by highlighting the regimes under which we can expect the proposed attacks to succeed *and fail*. In the latter case, results that highlight when the proposed attacks fail would point to potential defenses. If, for example, attacks would fail under certain conditions, this would point to key characteristics that are required to make use of recourse or counterfactual explanations safely.

**Justification For Why Not Higher Score:**

See review.

**Justification For Why Not Lower Score:**

N/A

---

### Decision · Program_Chairs · 2024-01-16

Reject